# ToolComp: A Multi-Tool Reasoning & Process Supervision Benchmark

## Abstract

Despite recent advances in AI, the development of systems capable of executing complex, multi-step reasoning tasks involving multiple tools remains a significant challenge. Current benchmarks fall short in capturing the real-world complexity of tool-use reasoning, where verifying the correctness of not only the final answer but also the intermediate steps is important for evaluation, development, and identifying failures during inference time. To bridge this gap, we introduce ToolComp, a comprehensive benchmark designed to evaluate multi-step tool-use reasoning. ToolComp is developed through a collaboration between models and human annotators, featuring human-edited/verified prompts, final answers, and process supervision labels, allowing for the evaluation of both final outcomes and intermediate reasoning. Evaluation across six different model families demonstrates the challenging nature of our dataset, with the majority of models achieving less than 50% accuracy. Additionally, we generate synthetic training data to compare the performance of outcome-supervised reward models (ORMs) with process-supervised reward models (PRMs) to assess their ability to improve complex tool-use reasoning as evaluated by ToolComp. Our results show that PRMs generalize significantly better than ORMs, achieving a 19% and 11% improvement in rank@1 accuracy for ranking base and fine-tuned model trajectories, respectively. These findings highlight the critical role of process supervision in both the evaluation and training of AI models, paving the way for more robust and capable systems in complex, multi-step tool-use tasks. [*]

## 1 Introduction

Recent advancements in large language models (LLMs) have demonstrated remarkable progress in a range of natural language processing tasks. These models have achieved state-of-the-art performance across diverse benchmarks, including question answering, summarization, and reasoning tasks. In order to further increase the usefulness of LLMs, a growing area of research is centered around the development of agentic capabilities, particularly their ability to autonomously interact with external tools to solve complex, multi-step tasks as well as to interact with human systems such as the web or mobile devices.

However, evaluating the effectiveness of these tool-use capabilities remains a pressing challenge. While there have been notable efforts in developing benchmarks for tool-use capability, these often assess isolated instances of tool use, focusing on whether the model can invoke the correct tool at the right time (Huang et al., 2024; Zhuang et al., 2023; Peng et al., 2021). Additionally, while benchmarks for multi-step tool usage exist, most focus only on scoring the correctness of the final answer (Mialon et al., 2023), despite that the complex nature of multi-step reasoning often requires the evaluation for partial correctness or step-wise correctness of the reasoning trajectories. This can be valuable for both understanding model failure modes and developing systems that can improve upon these intermediate reasoning flaws.

To address these shortcomings, we introduce ToolComp, a benchmark comprising 485 complex, human-verified prompts that require language models to chain together multiple tool calls, accompanied by human-edited step-wise and final answers. By demanding intricate tool interactions and

---

[*]Code, ToolComp benchmark, and synthetic training data will be made publicly available.

providing human verification, ToolComp offers a rigorous assessment of a model's ability to perform complex, multi-step reasoning and tool use. We evaluate the current landscape of state-of-the-art models on their ability to chain together tool calls to reach the final answer, as well as their step-wise reasoning ability.

Moreover, in light of recent works demonstrating how process supervision significantly improve reasoning in language models (Lightman et al., 2023; Wang et al., 2024a), we explore the best methods for improving agentic tool-use reasoning by conducting an initial comparative analysis between process-supervised reward models (PRMs) and outcome-supervised reward models (ORMs) on ToolComp. Our results demonstrate that process-supervised models outperform outcome-based approaches, underscoring the importance of training models with and evaluating against process supervision signals.

In order to avoid early contamination, we plan to open source the entire benchmark when either 1) any model scores over 85% on ToolComp or 2) at the end of 2025, whichever comes earlier.

## 1.1 CONTRIBUTIONS AND KEY TAKEAWAYS

Our key contributions and takeaways are summarized as follows:

- **Introduction of ToolComp** We introduce ToolComp, a multi-tool reasoning and process supervision benchmark with 485 human-edited/verified prompts and final answers, designed to evaluate a model's ability to perform multi-step tool-use tasks **(Section 3)**.

- **Step-by-Step Process Annotations** ToolComp includes 1731 detailed per-step supervision labels, enabling a comprehensive assessment of a model's intermediate reasoning when performing complex, multi-step tool-use tasks **(Section 3)**.

- **Assessment of State-of-the-Art Models** We evaluate 16 models across 6 different model families on their ability to perform complex multi-step tool-use tasks as well as their intermediate reasoning ability. We find that o1-preview has the best performance, achieving 61.83% against the human-verified final answers and 80.18% against the process supervision labels **(Section 4 and Section A)**.

- **Process-Supervision Outperforms Outcome-Supervision** Our analysis shows that process-supervised reward models outperform outcome-based reward models by 19% in rank@1 accuracy on base model generations and by 11% in rank@1 accuracy on fine-tuned model generations **(Section 5 and Section B)**.

- **Release of Synthetic Training Data and Evaluation Pipeline** We provide the complete synthetic training dataset, which includes 11K prompts, 11K preferred and dis-preferred full trajectories for ORM training, and 15K preferred and dis-preferred steps for PRM training. We also release the evaluation pipeline for ToolComp. **(Section 5)**.

## 2 RELATED WORKS

**Benchmarks for Complex Tool Use Planning** With rising interest in tool-augmented LLMs (Schick et al., 2023; Patil et al., 2023; Qin et al., 2023), several benchmarks have been introduced to assess their abilities. Earlier benchmarks were designed to assess a model's ability to do proper retrieval, execution, and extraction of one tool call for specific tasks such as general question answering (Yang et al., 2018; Joshi et al., 2017), fact verification (Thorne et al., 2018), or answering temporal queries (Chen et al., 2021; Kasai et al., 2024; Zhang & Choi, 2021; Dhingra et al., 2022; Vu et al., 2023). However, these benchmarks fail to assess a model's ability to plan and chain together multiple tool calls to answer more complex queries. More recent benchmarks aimed at evaluating multiple tool calls are often placed within or dependent on state-full systems (such as a code-base and/or a dynamic database) (Yan et al., 2024; Jimenez et al., 2024; Liu et al., 2023). Although these types of benchmarks assess a language model's ability to chain together multiple tool calls, the evaluation may penalize general-purpose language models that are not familiar with the given environments. Other benchmarks primarily rely on state-based evaluations, where the final state of the system is assessed against the desired state (Li et al., 2023; Peng et al., 2021), or win-rates against another reference state-of-the-art model (Qin et al., 2023), both of which lack the rigour of human-verified ground truth final answers. Closest to our work, the GAIA benchmark is a collection

Table 1: The contributions and metadata of popular benchmarks in Tool Use. Our work, ToolComp, is shown in the first column. From left to right, we include work from Mialon et al. (2023), Yan et al. (2024), Qin et al. (2023), Li et al. (2023),and Xu et al. (2023). * Although 2 of the 8 tools are not evaluated by simply matching a verified final answer, the remaining 6 have verified final answers.

| Resource | ToolComp | GAIA | BFCL | ToolBench | API-Bank | ToolBench |
|---|---|---|---|---|---|---|
| Real-World API Calls | ✓ | ✓ | ✓ | ✓ | ✓ | ✓ |
| Multi-Tools Scenario | ✓ | ✓ | ✓ | ✓ | ✗ | ✗ |
| Multi-Step Reasoning | ✓ | ✓ | ✓ | ✓ | ✓ | ✓ |
| Step-Wise Labels | ✓ | ✗ | ✗ | ✗ | ✗ | ✗ |
| Verified Final Answer | ✓ | ✓ | ✗ | ✗ | ✗ | ✓* |
| Number of Tools | 11 | 23 | 3 | 3451 | 53 | 8 |

of complex tool-use queries that require multi-step tool-use reasoning and associated ground-truth answers (Mialon et al., 2023). Crucially, it does not contain step-wise labels, which can be important for identifying where an error occurred and providing precise feedback. Additionally, a significant portion of GAIA requires specialized capabilities such as web browsing, multi-modality, and diverse file-type reading. In our work, we focus on text-only tasks in order to disentangle specialized capabilities and multi-step reasoning, allowing us to focus on the latter.

**Process Reward Models** Recent work has shown the power of utilizing process supervision signals, which are granular signals on the step-wise correctness of a solution, as opposed to outcome supervision signals, which are broad signals on the correctness of the entire solution. Utilizing these signals, Lightman et al. (2023) and Wang et al. (2024a) have shown dramatic improvements in performance in ranking solutions to mathematical reasoning tasks and using these signals to further improve performance in traditional RLHF algorithms such as Proximal Policy Optimization (PPO) (Schulman et al., 2017).

In this work, through a hybrid human-AI annotation workflow, we generate per-step process supervision labels, which uniquely enable us to rigorously evaluate a model's intermediate reasoning capability. Table 1 provides a comparative overview of popular tool-use benchmarks, including our work, ToolComp. In addition, we investigate how to best apply process supervision signals to improve multi-step tool-use reasoning, which introduces novel design challenges compared to its application in mathematical reasoning. For instance, the granularity of supervision becomes a key consideration, where we must decide between supervising the entire ReAct (Yao et al., 2023) process or its subcomponents. These design choices alongside comparisons with outcome supervision are explored in detail in Section 5.

## 3 TOOLCOMP

### 3.1 TOOLS

For the creation of this benchmark and evaluation framework, we support 11 tools: Date, Current Weather, Historical Weather (Zippenfenig, 2024), Calculator, Wiki Search (Majlis, 2017), Google Search (SerpApi, 2024), Wolfram Alpha (Wolfram Research, 2024), Intra-day Stock Info, Daily Stock Info, Stock Symbol Search (AlphaVantage), and Python. There were several considerations when choosing these set of tools, namely, we wanted to cover a broad range of use cases from fact retrieval to financial assistant, have some overlap in use cases to encourage various valid trajectories, ensure the tools are general enough to not require specialized knowledge for LLMs to use, and allow for interesting interactions between tools. A detailed breakdown of each tool, including descriptions, parameters, input examples, and output examples are available in Appendix G.

### 3.2 REACT FORMAT

We chose the ReAct format as it is frequently used for tool use and agentic workflows (Wang et al., 2024b; Mekala et al., 2024; Zhuang et al., 2023). The ReAct format combines reasoning and tool calls by prompting the model to first generate a thought, which contains the rationale behind the

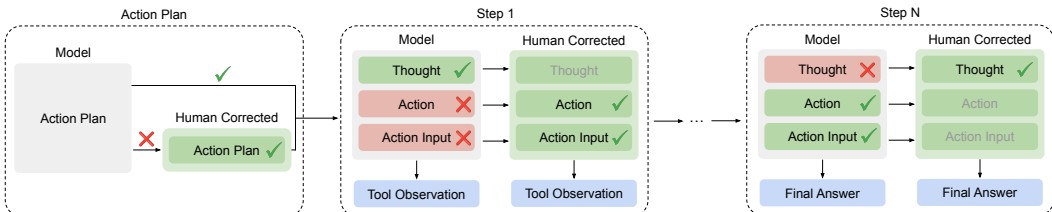

Figure 1: An example annotation path for collecting data that provides tool-call trajectories with human verified-final answers along with step-by-step process supervision labels. Each model generated step (Action Plan and ReAct steps) are first labelled as correct or incorrect. For the components labelled incorrect, a rewrite is made to correct the corresponding component. The annotations and rewrites are made by human annotators for the benchmark (or by a state-of-the-art LM for generating synthetic training data as further described in Section 5.1). A full annotated trajectory example is available in Appendix F.2.

following tool call action (Yao et al., 2023). The structured nature of the ReAct format into a thought, action, action input, and observation allows us to collect granular signals at each sub-step, and the relative simplicity of the ReAct format makes it easier to operationalize for annotations.

### 3.3 PROMPT CREATION

In developing the prompts for this dataset, there are two main criteria we desire each prompt to satisfy: 1) the solution to the prompt contains a chain of dependent tool calls to answer and 2) the final answer to the prompt can be programmatically verified. To achieve this, we generate a set of candidate prompts through few-shot prompting which are then refined and validated by human annotators. The overall process includes 1) manually developing in-context (IC) examples, 2) generating initial prompts, 3) an iterative process of filtering prompts, adding filtered prompts as negative IC examples, and regenerating more prompts, and 4) human refinement. These steps are described in more detail in Appendix C.1

### 3.4 CHAT VS. ENTERPRISE USE CASES

In creating the benchmark, we developed two subsets of prompts, coined ToolComp-Enterprise and ToolComp-Chat. ToolComp-Enterprise allows the use of 11 tools and aims to emulate settings in which LLM agents must compose a larger number of expressive APIs together correctly, such as in enterprise settings. The second subset, ToolComp-Chat, is designed to test general purpose chatbots with the minimally sufficient set of tools for information retrieval and processing tasks, namely Google Search and Python. We chose only google search and python execution as these are standard tools found in major chat-bot providers. We only allow the respective tools for each subset during prompt generation, labeling, and evaluation. ToolComp-Enterprise contains 287 examples and ToolComp-Chat contains 198 examples.

### 3.5 LABEL CREATION

To create the process supervision labels as well as the final answer for each prompt, we utilize a hybrid human-AI approach, where the language model and human annotators use the same tools to collaborate to get to the final answer. We start by prompting the Policy Model LLM to outline a plan, called Action Plan, on which tools to call and in what order using the prompt in E.1. We have human annotators validate/modify the Action Plan, which is then appended to the sequence before using the LLM to formulate tool calls. We then use the LLM to call tools in the ReAct format, where the specific prompt can be found in E.2.

We asked the annotators to rate if a step is Correct (i.e., the step is a reasonable action towards achieving the final answer) or Incorrect (i.e., the step is nonsensical, incorrect, or is not a reasonable action towards acheiving the final answer). All components of the ReAct Step (Thought, Action, Action Input) must be marked as Correct or Incorrect by the annotator. If the annotator marks a step as Correct, the model is allowed to proceed further and generate the next step. If the annotator

deems a step as Incorrect, they must modify the step to make it correct. Once corrected, the model is then prompted to advance to the next step with the human-corrected step as part of its context. This is repeated until the Finish Action is chosen by the LLM and marked as Correct by the annotator or until the annotator corrects an Action step to 'Finish' because we have enough information to answer the question. The overall flow is shown in Figure 1. An example golden trajectory is available in Appendix F.1 and an example annotated trajectory is available in Appendix F.2. We use FireFunction-V1 as the Policy Model LLM (at the time, this was the best open-source tool-use LLM) and humans as the annotators (Fireworks, 2024).

With this process, we retrieve, per task, a valid step-by-step chain of tool calls that successfully gets to the final answer along with step-wise correct/incorrect labels and associated rewrites. The correct/incorrect labels and the associated rewrites allow us to assess intermediate reasoning through LLM-as-judge evaluations (described in Section 4.3).

### 3.6 QUALITY CONTROL

To ensure the highest quality of ToolComp, we conduct a thorough manual inspection of all examples. Any data samples with ambiguous prompts, erroneous process supervision labels, or incorrect final answers are redone. After the initial creation of the benchmark, the authors collaborated with three trusted annotators to perform a final re-review of all samples and make any necessary corrections.

As a final quality control step, we evaluate the entire benchmark using GPT-4o (May 2024), GPT-4 Turbo, Claude 3.5 Sonnet, and Llama 3.1 405b (OpenAI et al., 2024; Dubey et al., 2024; Anthropic). We identify the set of data samples where all models' answers differed from the ground truth final answers. We then repeated the refinement process on these samples, as they represented the most challenging and/or potentially mislabeled data points. This iterative approach yielded the final version of ToolComp.

## 4 TOOLCOMP EVALUATIONS

### 4.1 EVALUATION METRIC

We have two metrics to evaluate the quality or the correctness of a model's final answers: LLM Grading and Exact Match. For the final answer evaluations in this section (Table 2), we use LLM Grading since it rewards correct answers without penalizing minor formatting issues. Our Exact Match evaluation methodology and the corresponding results are shown in Appendix A.1.

**LLM Grading**    By using LLM grading against ground truth answers we opt to be charitable to exact formatting and focus on assessing the tool use capabilities of the model. We intentionally choose not to focus on final answer formatting given that (1) there are existing benchmarks that assess formatting ability (e.g. FOFO (Xia et al., 2024)) and (2) our final answers are quite complex, containing multiple elements, lists which may or may not be sorted, and dictionaries. This approach prompts an LLM Judge to look at the prompt, the ground truth answer, and the model's answer and asks the model to classify it as Incorrect, Correct, or Correct with Bad Formatting. We use GPT-4 Turbo as the de-facto judge for all of our models (OpenAI et al., 2024). The prompt used is shown in Appendix E.5. We consider both Correct and Correct with Bad Formatting as a win (accurate) and Incorrect as a loss (inaccurate).

### 4.2 FINAL ANSWER EVALUATIONS

The overall scores of the various state-of-the-art tool-use models are shown in Table 2. We combine ToolComp-Chat and ToolComp Enterprise subsets to get an overall score and 95% confidence-intervals (CIs) for the entire benchmark. We use native function calling for all the models except for o1-preview. Since the o1-preview API does not accept system prompts nor allows for native function calling, we prepend our ReAct System Instruction (Appendix E.2) to the user query. Additionally, we allow each model to retry up to 3 times if it fails to output a final answer. This is determined by whether there is a parse-able JSON object in the final output with the key "final_answer". To ensure scores are not indicative of tool or endpoint failures due to rate limiting, we use verbose logging to

Table 2: Accuracy and the 95% CIs of all selected models using the final answer and scored using an LLM judge (Dubey et al., 2024; OpenAI et al., 2024; Gemini et al., 2024; Anthropic; Mistral; Cohere). We combined the results of each subset to give an overall score for the entire benchmark. Exact Match results are reported in Appendix A.1 but the rankings do not significantly differ, with the top 5 and bottom 4 models remaining the same. *Llama models sometimes fail to call tools/terminate early or call tools in the wrong format. Using constrained decoding and other techniques to guarantee structured outputs can improve their performance. **Since o1-preview does not support native function calling via API, we prompt the model to formulate tool calls in the ReAct format.

| Model Family | Model Name | Total (%) | Chat (%) | Enterprise (%) |
|---|---|---|---|---|
| OpenAI | o1-preview** | $61.83 \pm 4.34$ | $55.1 \pm 6.96$ | $66.43 \pm 5.47$ |
| | GPT-4o (Aug 2024) | $58.68 \pm 4.39$ | $56.85 \pm 6.92$ | $59.93 \pm 5.67$ |
| | GPT-4o (May 2024) | $58.44 \pm 4.38$ | $49.5 \pm 6.96$ | $64.58 \pm 5.52$ |
| | GPT-4 Turbo Preview | $57.61 \pm 4.39$ | $53.03 \pm 6.95$ | $60.76 \pm 5.64$ |
| | GPT-4 | $45.89 \pm 4.43$ | $37.88 \pm 6.78$ | $51.39 \pm 5.77$ |
| | GPT-4o Mini | $44.03 \pm 4.41$ | $32.83 \pm 6.54$ | $51.74 \pm 5.77$ |
| Anthropic | Claude 3.5 Sonnet | $58.03 \pm 4.39$ | $56.06 \pm 6.91$ | $59.38 \pm 5.67$ |
| | Claude 3 Opus | $51.03 \pm 4.44$ | $48.49 \pm 6.96$ | $52.78 \pm 5.77$ |
| | Claude 3 Sonnet | $48.56 \pm 4.44$ | $40.4 \pm 6.84$ | $54.17 \pm 5.78$ |
| Google | Gemini 1.5 Pro (Aug 2024) | $56.61 \pm 4.41$ | $51.27 \pm 6.98$ | $60.28 \pm 5.66$ |
| | Gemini 1.5 Pro (May 2024) | $38.43 \pm 4.34$ | $35.5 \pm 6.57$ | $40.42 \pm 5.68$ |
| Mistral | Mistral Large 2 | $46.30 \pm 4.43$ | $40.4 \pm 6.84$ | $50.35 \pm 5.78$ |
| Meta | Llama 3.1 405B Instruct* | $46.19 \pm 4.44$ | $40.1 \pm 6.84$ | $50.35 \pm 5.78$ |
| | Llama 3.1 70B Instruct* | $35.74 \pm 4.27$ | $33.5 \pm 6.59$ | $37.23 \pm 5.6$ |
| | Llama 3.1 8B Instruct* | $12.81 \pm 2.98$ | $6.09 \pm 3.34$ | $17.42 \pm 4.39$ |
| Cohere | Command R+ | $26.13 \pm 3.91$ | $20.2 \pm 5.59$ | $30.21 \pm 5.3$ |
| | **Average** | $\mathbf{46.64 \pm 4.27}$ | $\mathbf{41.08 \pm 6.50}$ | $\mathbf{50.46 \pm 5.58}$ |

Table 3: Accuracy and the 95% CIs (third column) of all of our models on the process supervision labels in ToolComp. We evaluate the model's effectiveness as a pairwise judge in selecting the human-corrected answer versus the model-generated incorrect answer. We show judge accuracy using the ReAct steps (fourth column) and the Action Plan (fifth column).

| Model Family | Model Name | Total (%) | ReAct (%) | Action Plan (%) |
|---|---|---|---|---|
| OpenAI | o1-preview | $80.19 \pm 1.89$ | $79.62 \pm 2.22$ | $81.76 \pm 3.55$ |
| | GPT-4o (Aug 2024) | $72.61 \pm 2.11$ | $72.84 \pm 2.46$ | $71.98 \pm 4.13$ |
| | GPT-4o (May 2024) | $71.24 \pm 2.14$ | $71.37 \pm 2.49$ | $70.88 \pm 4.17$ |
| | GPT-4 Turbo Preview | $70.66 \pm 2.15$ | $70.18 \pm 2.52$ | $71.98 \pm 4.13$ |
| | GPT-4o Mini | $63.02 \pm 2.28$ | $64.27 \pm 2.64$ | $59.56 \pm 4.51$ |
| | GPT-4 | $60.02 \pm 2.32$ | $55.87 \pm 2.74$ | $71.54 \pm 4.15$ |
| Anthropic | Claude 3.5 Sonnet | $66.46 \pm 2.23$ | $67.74 \pm 2.58$ | $62.97 \pm 4.44$ |
| | Claude 3 Opus | $64.28 \pm 2.27$ | $64.55 \pm 2.64$ | $63.52 \pm 4.42$ |
| | Claude 3 Sonnet | $61.10 \pm 2.31$ | $62.93 \pm 2.67$ | $56.04 \pm 4.56$ |
| Google | Gemini 1.5 Pro (Aug 2024) | $69.11 \pm 2.19$ | $68.48 \pm 2.56$ | $70.88 \pm 4.17$ |
| | Gemini 1.5 Pro (May 2024) | $67.89 \pm 2.21$ | $67.72 \pm 2.58$ | $68.35 \pm 4.27$ |
| Mistral | Mistral Large 2 | $72.67 \pm 2.11$ | $73.16 \pm 2.45$ | $71.32 \pm 4.16$ |
| Meta | Llama 3.1 405B Instruct | $71.62 \pm 2.13$ | $73.87 \pm 2.42$ | $65.39 \pm 4.37$ |
| | Llama 3.1 70B Instruct | $70.75 \pm 2.15$ | $71.33 \pm 2.50$ | $69.12 \pm 4.25$ |
| | Llama 3.1 8B Instruct | $57.63 \pm 2.34$ | $59.60 \pm 2.71$ | $52.20 \pm 4.56$ |
| Cohere | Command R+ | $61.31 \pm 2.30$ | $64.91 \pm 2.63$ | $51.32 \pm 4.59$ |
| | **Average** | $\mathbf{67.54 \pm 2.20}$ | $\mathbf{68.03 \pm 2.55}$ | $\mathbf{66.18 \pm 4.28}$ |

log all failures and retry any prompt where a tool or model outputs failed due to rate/load limits. In addition, we run error analysis on the types of failures for each model. A description of the error category taxonomy and the breakdown of failure modes for each model can be found in Appendix A.2.

We also show exact match evaluation numbers in Table 6 of Appendix A.1 to ensure that our LLM Judge (GPT-4 Turbo) isn't biased in favor of outputs from the same model family. Upon inspection of the discrepancies (i.e., examples marked correct by the LLM judge but incorrect under exact match), we find that they are all due to issues with the model's formatting of the final answer despite getting to the correct answer.

### 4.3 LLM-as-Judge Evaluations

We further evaluate these models using our process supervision labels, aiming to assess each model's effectiveness as a pairwise judge in selecting the human-corrected step over the step generated by the original policy used during annotation. To mitigate position bias, we swap the order of the human-corrected and model-generated steps and conduct two separate predictions for each arrangement. Additionally, models are permitted to indicate a tie. If a model designates a tie at least once, or consistently predicts the same position (before and after swapping) for a given data sample, we classify the outcome as a tie. Mirroring the methodology used in RewardBench (Lambert et al. (2024)), we score losses as 0, ties as 0.5, and wins as 1. We show the results below in Table 3.

### 4.4 Intermediate Reasoning vs. Final Answer

Figure 2 shows the correlation between a model's intermediate reasoning performance and final answer accuracy based on the multi-step tool-use tasks in ToolComp. The standard Pearson correlation coefficient is $r = 0.63$ with a statistical $p$-value of $0.0084$, which makes the correlation statistically significant under a significance level of $0.05$ (Freedman et al., 2007). Intuitively, this suggests that with stronger step-wise performance as assessed by our LLM-as-judge evaluations, we can expect an increased likelihood of reaching the correct final answer. However, the moderate magnitude of the correlation value could be due to additional signals captured by the step-wise reasoning evaluations that are not captured by evaluating final answers. Work done by Havrilla et al. (2024) similarly suggests that there is complementary and non-overlapping information in step-wise and final answer refinement, further highlighting the importance of assessing intermediate reasoning.

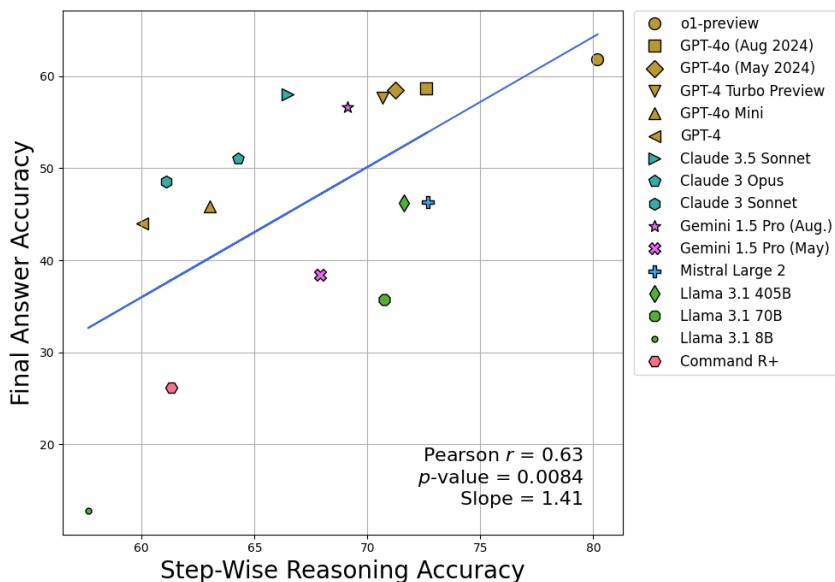

Figure 2: Comparison of step-wise reasoning accuracy (x-axis) and final answer accuracy (y-axis) on ToolComp across 6 different model families.

# 5 PROCESS SUPERVISION VS. OUTCOME SUPERVISION

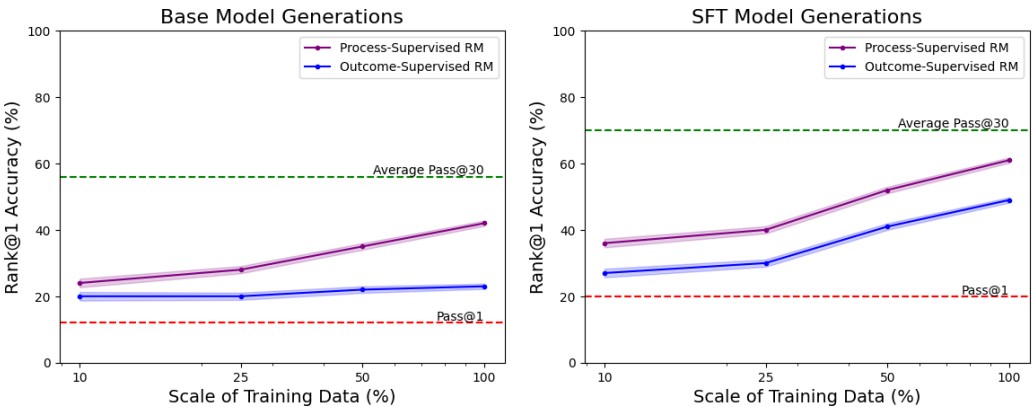

Figure 3: A comparison of outcome-supervised and process-supervised reward models across various scales of training data (10%, 25%, 50%, 100%), evaluated by their ability to pick out the best answer out of 30 tool call trajectories. The 95% confidence intervals captures the variance of 500 random samples of 30 completions out of 50 completions. We plot both the performance on generations from Llama-3.1-8b-Instruct (left) and Llama-3.1-8b-Instruct fine-tuned on all the preferred trajectories (right) (Dubey et al., 2024). The plot also shows the Pass@1 given by greedy sampling and the average Pass@30 accuracies for the respective generating models.

Recent works have demonstrated the power of process supervision signals in domains such as mathematical reasoning (Lightman et al., 2023; Wang et al., 2024a). Despite this, the application of process-supervised signals towards tool-augmented LLMs remains under-explored. By focusing on the process rather than just the outcome, process-supervised models could provide more granular feedback during multi-step tool-use, leading to faster convergence, especially in complex applications where each step is associated with dynamic feedback from the environment. In this section, we provide a preliminary analysis to assess the value of process supervision for multi-step tool-use by training PRM and ORM models using additionally generated synthetic data.

## 5.1 EXPERIMENT DESIGN

**Training Dataset** For the generation of synthetic prompts and the process/outcome supervision labels, we mirror the strategies outlined in Section 3.3 and Section 3.5, respectively. We use Llama-3.1-8b-Instruct as the Policy Model generator and Chat GPT-4o as the Critic Model generator (OpenAI et al., 2024; Dubey et al., 2024). A detailed accounting of the training data generation, including generation parameters, dataset sizes, and methodology can be found in Appendix D.1. Furthermore, the construction of the outcome-supervised, process-supervised reward modelling and supervised fine-tuning datasets are detailed in Appendix D.2.

**Reward Model Training Objective** We equip the base model, Llama-3.1-8b-Instruct (Dubey et al., 2024), with a linear layer to serve as the reward head. For ORM training, the reward model places a probability of correctness of the whole preferred and dis-preferred trajectory. For PRM training, we experiment with 4 different levels of process supervision. The first axis of variation experiments with including/excluding the "Observation" of the ReAct step, which is the output observed from the tool call. The second axis of variation ablates the granularity of the process signals, i.e., rewarding the whole ReAct step or rewarding each substep of the ReAct step (recall that a ReAct step contains Thought, Action, and Action Input, the correctness of which is determined/modified by the Critic Model). This leads us to 4 total supervision methods: Full ReAct step with or without Observation, Sub ReAct Step with or without Observation. Under each type of supervision, the PRM places a probability of correctness of the preferred step and dis-preferred step. The training objective is given by the average Binary Cross Entropy loss, assessing the probability of correctness with the corresponding label. A more detailed explanation of the training objective is in Appendix D.3 and training implementation along with hyper-parameters is in Appendix D.5.

**Evaluation**   To evaluate, we use our ORM and PRM models to select the best response out of a set of candidates. Specifically, we use a generator model to produce 50 completions per problem in ToolComp. We experiment with two different generator model in these experiments: base Llama-3.1-8b-Instruct and a supervised fine-tuned Llama-3.1-8b-Instruct model, which is trained on all of the full preferred trajectories in the synthetic training data. After collecting these completions, we use the trained ORM and PRM to rank these completions, returning the best answer according to this ranking. We then assess the rank@1 accuracy by judging the correctness of said highest scoring completion against the ground truth final answer per problem. For the ORM, we simply use the reward score it places on each completion. For the PRM, in order to compress per-step scores into one single score for the entire chain, we experiment with different aggregation functions, namely: min, max, average, and product.

To account for variance in the rankings, we perform 500 permutations of 30 random samples from the 50 total completion per problem, and calculate the average rank@1 accuracy. Moreover, we vary the dataset sizes at 10, 25, 50, and 100 percent of the full dataset in order to assess how performance of each method scales with increasing training data.

## 5.2   Results

**PRM outperforms ORM in selecting the best trajectory**   In Figure 3, we observe that for both the base model generations and the SFT model generations, our best PRM model (see Table 4) outperforms the ORM in rank@1 accuracy. On the base model generations, the PRM achieves an accuracy of 42.65% compared to the ORM accuracy of 23.89%. Moreover, towards enhancing an already fine-tuned model, we see that the PRM is able to push rank@1 accuracy to 60.25%, nearly matching the generator's pass@30 performance. Overall, these results suggest that PRMs are able to efficiently translate these per tool call step-level signals to provide superior performance than utilizing just outcome-level signals. In Appendix B, we also find the PRM scales better than the ORM on increasing prompt complexity.

**Full step with observation is the best PRM supervision method**   Table 4 shows the performance of the PRM using different supervision methods trained on the full-scale dataset. We see that providing the model signals about whether an entire step, including the observation, was correct or incorrect led to the best performance. These findings suggest that 1) intermediate information from the environment (tool call outputs) provide valuable signal to PRMs as these dynamic signals provide additional insight in determining the correctness of a trajectory, and 2) there is a balance to be struck when determining the granularity of supervision during training. Intuitively, providing less granular, full-step signals gives the model more freedom to learn and identify what makes one step better than another, without being constrained by potentially noisy or overly detailed sub-step labels. This allows the model to generalize more effectively, rather than being limited by specific, finer-grained supervision.

Table 4: Comparison of the average rank@1 accuracy across different methods of PRM supervised-training, which are all trained on the full-scale training dataset.

| Method | Rank@1 Accuracy (%) |
| --- | --- |
| Full Step with Observation | 60.25 |
| Full Step without Observation | 55.43 |
| Sub Step with Observation | 49.48 |
| Sub Step without Observation | 45.70 |

**PRM scales better than ORM with increasing data**   Part of the design of our experiment is to measure the scaling performance of PRMs versus ORMs as we incorporate more training data. In Figure 3, we find that the PRM rank@1 performance consistently outperforms the ORM at all training data scales in ranking both the base model completions as well as the SFT model completions. Interestingly, we observe greater performance scaling for the base model completions, emphasizing the importance of utilizing process level signals as the PRM is able to still pick out the best trajectory amongst lots of low-quality trajectories. This ability could also serve to pick out high-quality trajectories for further training the base model for multi-step tool-use reasoning.

Table 5: Comparison of the performance of different aggregation methods used to combine stepwise level PRM scores. Results here use the PRM model trained on all data with the Full Step with Observation supervision method.

| Method | rank@1 Accuracy (%) |
|---|---|
| Max | 60.25 |
| Min | 23.68 |
| Average | 25.74 |
| Product | 23.06 |

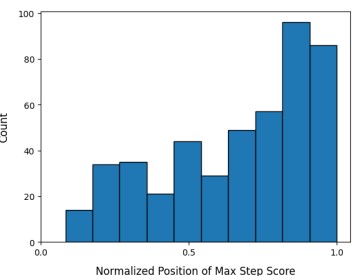

Figure 4: Distribution of the position of the maximum scoring step, normalized by the length of the trajectory, for the rank@1 selected trajectories.

**Max is the best PRM aggregation function**   Since the PRM provides a score for each step in the trajectory, an important design decision is how to combine step-level scores into a single score that can be used for ranking. Table 5 clearly demonstrates that max is by far the best aggregation function for scoring trajectories. By examining many trajectories and their step-level scores, we find that min and average both heavily penalize any wrong steps that are taken, even if the model eventually recovers and gets to the correct final answer. When using product as the aggregation function, the final aggregation results in low magnitudes that are biased towards shorter trajectories. Max is a better aggregation function because it avoids these aforementioned pitfalls and tends to favor the later steps (as shown by Figure 4) which are a better proxy for a successful trajectory.

## 6    LIMITATIONS AND FUTURE DIRECTIONS

In this study, we focus solely on applying outcome and process supervision to the reward model. Although fine-tuning the policy model with supervision from the reward model using reinforcement learning (RL) is a logical next step, we leave this for future work and focus instead on the contributed dataset and the value of process supervision even without RL.

A notable limitation of our work is the reliance on synthetic data to scale the policy. We hypothesize that incorporating human-generated data to expand the training set could enhance the tool-use capabilities beyond the performance of the state-of-the-art critic model, which was used to label our synthetic dataset.

Additionally, the restricted set of tools used in this work, primarily focused on information retrieval and data processing, presents another limitation. In contrast, a common approach in the field involves employing specialized models for various tasks such as image generation and translation. This opens up further questions regarding how process supervision could facilitate the scaling of more nuanced capabilities when integrating with other specialized models.

## 7    ETHICS STATEMENT

In the creation of this dataset and any further analysis, we abide by the ICLR Code of Ethics. We ensure all prompts in this dataset do not contain any harmful or sensitive material by requiring annotators to flag any such prompts. The authors of this paper have also manually inspected all the prompts and tool calls for harmful content. In addition, we applied best practices for code execution, ensuring that all the code execution is done in a sand-boxed environment for any past and/or future benchmark evaluations. We also ensured that all tools used have a permissive license for research purposes, and we plan to open-source both the code for running evaluations and the full benchmark dataset.

## 8 REPRODUCIBILITY

For the creation of the benchmark, we detail the exact process by which we create the dataset in Section 3. We also detail the exact evaluation method used to evaluate each model in Section D.4 and Appendix A.1. Moreover, for the training of the ORM, PRM and SFT models, we detail the exact process (including methodology, hyperparmeters, and additional settings) in Section 5.1 and Appendix D. We plan to open source both the code for evaluation and the benchmark dataset.

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

# A TOOLCOMP EXTENDED EVALUATIONS

In this appendix section, we include additional evaluations, namely the exact match grading (A.1) and error analysis for each model (A.2 and A.3).

## A.1 EXACT MATCH

This paradigm aims to assess both the tool use capabilities and the instruction/format following capabilities of the model. Formatting is particularly important when we want to use the LLM to automate a backend process. This paradigm programmatically evaluates unsorted lists (eg. prompt asks for a list of all states in the US), sorted lists (eg. prompt asks for a list of all states in the US in alphabetical order), numbers (eg. prompt asks for the areas of Texas in square miles) and strings (eg. prompt asks for the name of the football team that won the Superbowl in 2016)

Unsorted lists are sorted and exact matched (set match gets rid of duplicates) Sorted lists are exact matched Number are checked if they are within a tolerance param (the tolerance param is to account for variance among different sources online) String are stripped, lower cased, and exact matched

Table 6: Model Family Performance Comparison: Accuracy and 95% Confidence Intervals

| Model Family | Model Name | Total Accuracy (%) |
|---|---|---|
| OpenAI | o1-preview | $38.92 \pm 4.36$ |
| | GPT-4o (Aug 2024) | $43.52 \pm 4.43$ |
| | GPT-4o (May 2024) | $40.60 \pm 4.38$ |
| | GPT-4 Turbo Preview | $40.11 \pm 4.39$ |
| | GPT-4 | $38.45 \pm 4.34$ |
| | GPT-4o Mini | $34.70 \pm 4.25$ |
| Anthropic | Claude 3.5 Sonnet | $42.92 \pm 4.42$ |
| | Claude 3 Opus | $36.96 \pm 4.43$ |
| | Claude 3 Sonnet | $33.58 \pm 4.21$ |
| Google | Gemini 1.5 Pro (August 27, 2024) | $43.22 \pm 4.43$ |
| | Gemini 1.5 Pro (May 2024) | $27.36 \pm 3.98$ |
| Mistral | Mistral Large 2 | $33.63 \pm 4.21$ |
| Meta | Llama 3.1 405B Instruct* | $33.10 \pm 4.20$ |
| | Llama 3.1 70B Instruct* | $26.19 \pm 3.93$ |
| | Llama 3.1 8B Instruct* | $11.75 \pm 2.88$ |
| Cohere | Command R+ | $0.00 \pm 0.00$ |

## A.2 FINAL ANSWER FAILURE ANALYSIS

In order to better understand the reasons behind each model's failures, we come up with an Error Taxonomy and use GPT-4 Turbo to categorize the reasoning behind each failure. We note that the error categories are not mutually exclusive. We inspect the individual failure cases predicted by GPT-4 Turbo and find that it is reasonably accurate. The different categories and their definitions are shown in Table 7 and the error counts for each model is shown in Figure 5.

Table 7: Common Error Category Taxonomy.

| Category | Description |
|---|---|
| Final Answer Missing Information | The model's trajectory got to the final answer however the final answer fails to answer all parts of the prompt. |
| Called Incorrect Tool | The model called irrelevant tools that lead it down the wrong direction. |
| Incorrect Tool Call Formatting | The model tried to call the relevant tool but consistently used the wrong formatting for the input arguments (e.g., wrong input format, didn't include a required argument). You can tell this is occurring if the tool call's result is an error message. |
| Terminated Early Unexpectedly | The model stopped short of reaching the final answer even though it should have kept proceeding. It is unclear why the model stopped early. |
| Hallucinated Information | The model either didn't call the relevant tool and just made up information or it called the relevant tool but didn't use its outputs in the next tool call or final answer properly (made up information afterwards). |
| Misunderstood Tool Info | The model called the relevant tool but misunderstood the information it gave back. |
| Repeatedly Calling Same Tool | The model called the same tool with the same arguments multiple times (even though it didn't have any errors) and didn't use the returned info to proceed to the next step or the final answer. |
| Action Plan Flawed | The Action Plan provided to the model in the user query was fundamentally flawed. |
| Miscellaneous | The reason for the error doesn't fit into any of the above categories. |

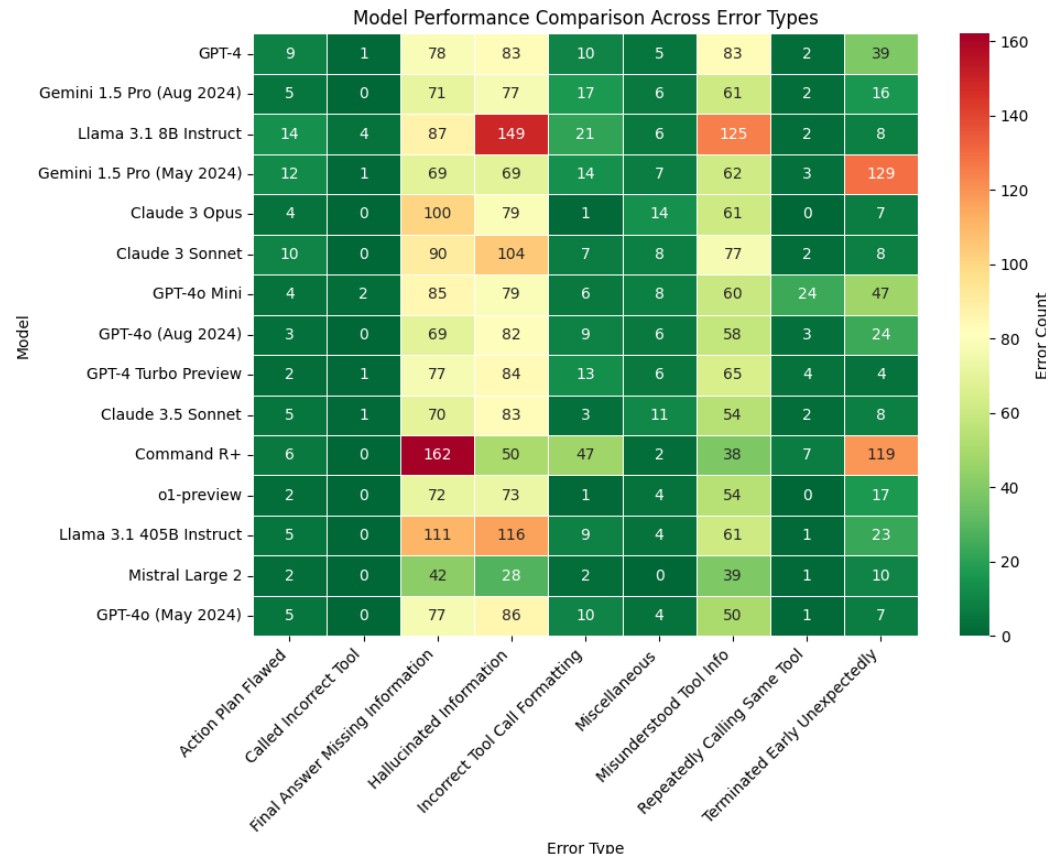

Figure 5: Breakdown of the various error categories in our taxonomy for each model (on the ToolComp-Enterprise).

### A.3 INTERMEDIATE REASONING FAILURE ANALYSIS

In this appendix section, we conduct a thorough failure analysis for the intermediate reasoning evaluations shown in Table 3.

#### A.3.1 REACT-STEP-ERROR-BASED FAILURE TRENDS IN MODELS

Figures 6 and 7 shows the count for type of mistake between the human corrected substep and the original incorrect substep whenever the model fails to pick the more appropriate trajectory (see Figure 1 for an overview on the annotation process). We define the failure cases in terms of which subset of the ReAct step needed correction. We end up with 5 different cases:

- **Case 1**: Thought Correct, Action Correct, Action Input Incorrect
- **Case 2**: Thought Incorrect, Action Incorrect, Action Input Incorrect
- **Case 3**: Thought Incorrect, Action Correct, Action Input Correct
- **Case 4**: Thought Incorrect, Action Correct, Action Input Incorrect
- **Case 5**: Thought Correct, Action Incorrect, Action Input Incorrect

Together, these figures highlight what types of errors are most common during a lapse in reasoning when picking the best next course of action or invoking a tool correctly. In particular, we notice that models often fail in reasoning about the better course of action when the deciding factor is in picking the better Action Input with all else equal.

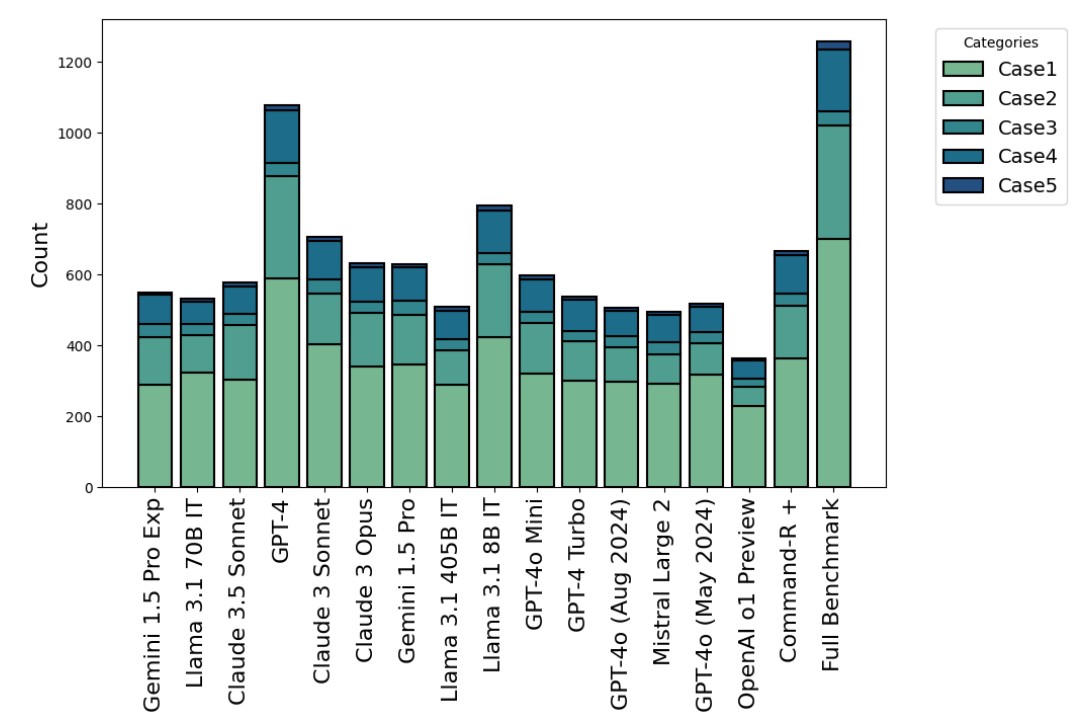

Figure 6: Histogram showing the LLM as judge evaluation failure counts for each model, which is further categorized by subset of the ReAct step that needed correction. Full Benchmark denotes the counts for the entire ToolComp benchmark. Recall from 4.3, we have 3 outcomes for LLM judge evaluation: win, tie, or loss. Here we count a failure as either a tie or a loss outcome.

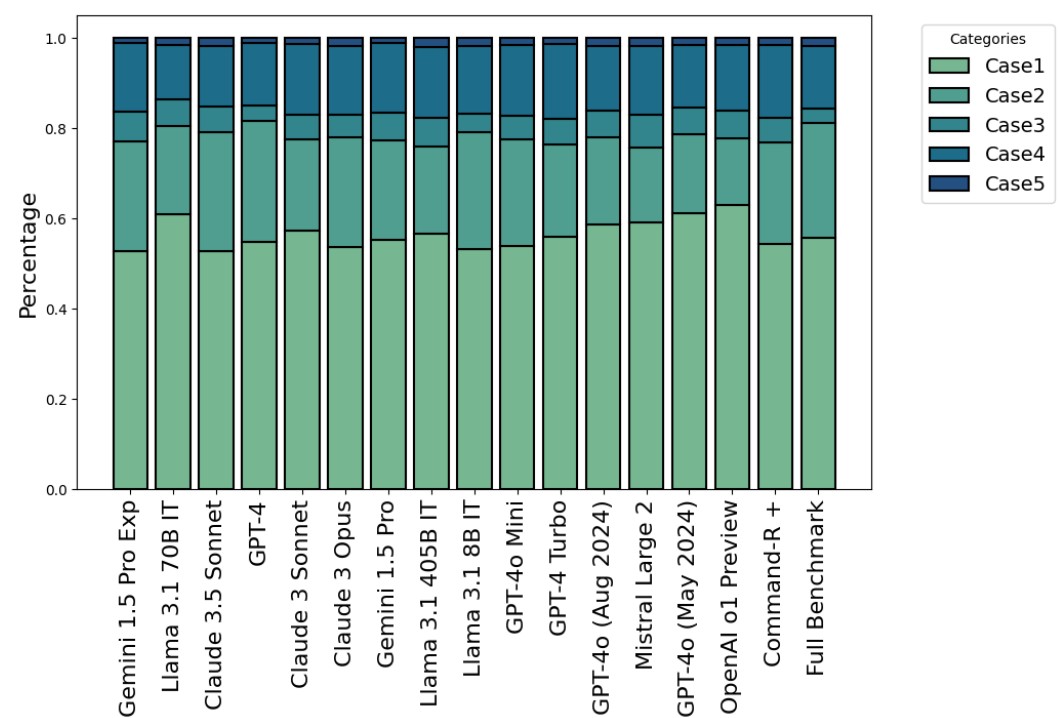

Figure 7: Density of the error-type between correct and incorrect step for the LLM as judge evaluation failures for each model. Full Benchmark denotes the distribution for the entire ToolComp benchmark.

### A.3.2 POSITION-BASED ERROR TRENDS IN MODELS

Figures 8 and 9 shows the count and percentage of the relative positions where each respective model failed to chose the better step when serving as an LLM judge choosing between two steps. In order to calculate the position, we divide the step number at which the decision is taking place by the total number of steps in the trajectory and multiply by 100. Hence, the position of a step will be a number between 0 and 100. We bin these position values by increments of 20. Overall, these figures illustrate that most, if not, all of the models struggle when judging steps towards the middle-end (position values between 60 and 80) of the trajectory. Intuitively this makes sense because this is likely where models have to compose the observations of previous tools into the input for the next tool call, which requires more nuanced and sophisticated reasoning.

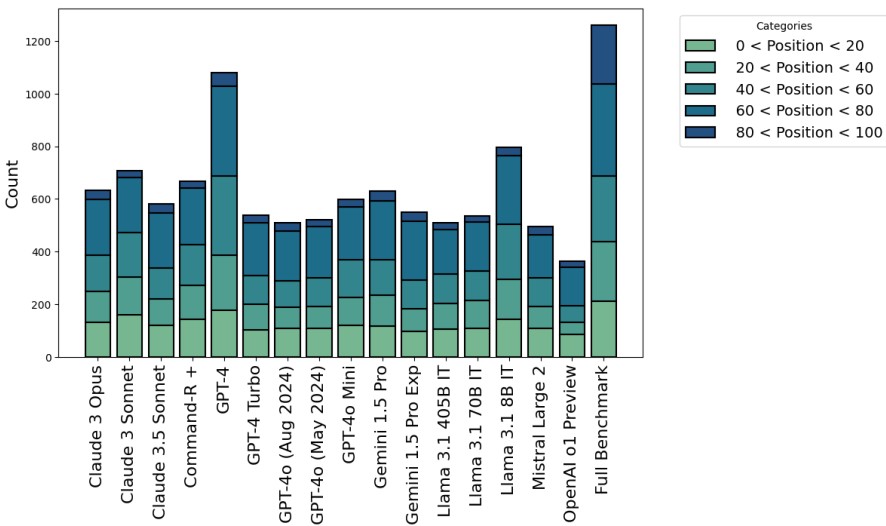

Figure 8: Histogram showing the LLM as judge evaluation failure counts for each model, which is further categorized by the position of the decision step. Full Benchmark denotes the counts for the entire ToolComp benchmark.

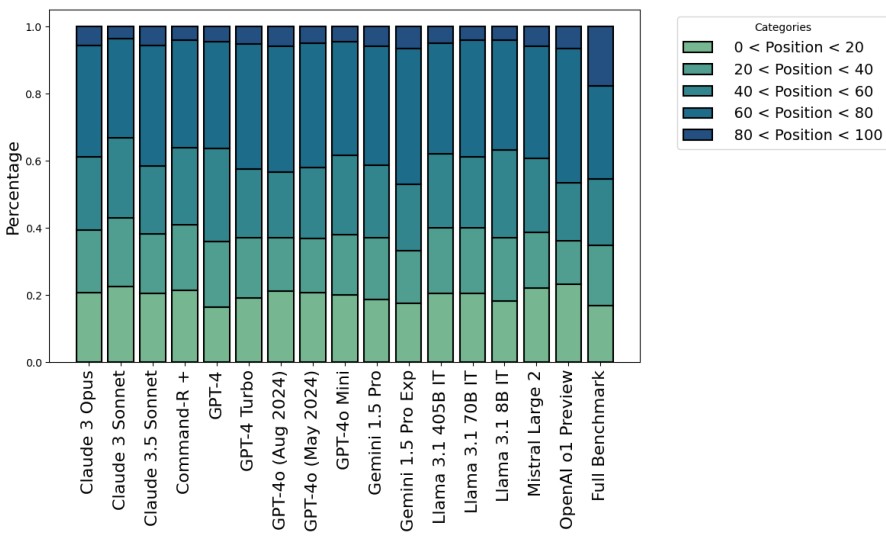

Figure 9: Density of the position of the LLM as judge evaluation failures for each model. Full Benchmark denotes the distribution for the entire ToolComp benchmark.

## B  PERFORMANCE SCALING ON INCREASING COMPLEXITY

In this appendix section, we compare how process supervised reward model performance scales with more complex tool use prompts.

**Categorizing Prompt Complexity**   We group ToolComp prompts into three categories of complexity — Easy, Medium, and Hard — based on the number of tool calls required in the human-verified trajectory (see Figure 1 for an overview on the annotation process) to answer the prompt:

- **Easy**: Prompts solved in 1–4 steps. (199 total prompts)
- **Medium**: Prompts solved in 5–8 steps. (210 total prompts)
- **Hard**: Prompts solved in 9–12 steps. (62 total prompts)

While there can be multiple valid trajectories of different lengths for the same prompt, we consider this categorization a reasonable proxy for complexity.

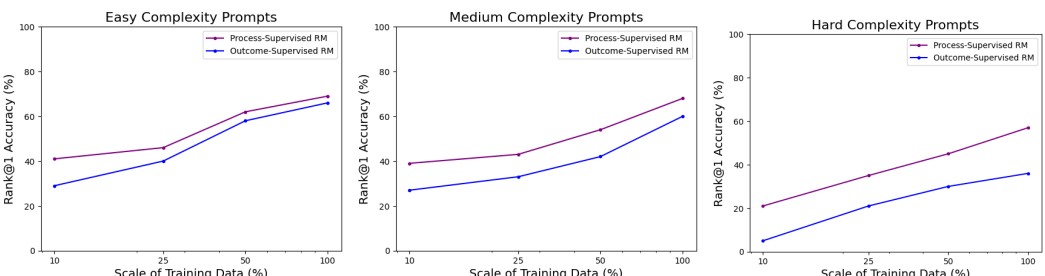

Figure 10: A comparison of outcome-supervised and process-supervised reward models across various scales of training data (10%, 25%, 50%, 100%) on different complexity of prompts. We use the same 50 completions from the fine-tuned Llama-3.1-8b-Instruct generator in the experiments in Section 5 and we consider all 50 completions when ranking the trajectories.

**PRM performance scaling is the greater for more complex prompts.**   Figure 10 compares the rank@1 performance scaling of ORM and PRM across the different prompt complexity. PRM consistently demonstrates better scalability for harder prompts, with the largest performance gains over ORM observed in the Hard category. This highlights that PRMs are particularly effective in handling more complex queries requiring sophisticated reasoning across multiple tool call steps.

## C  ToolComp Details

In this appendix section, we provide further details regarding benchmark creation steps such as prompt creation (C.1, C.2, C.3). We also provide additional benchmark metadata revolving different characteristics and statistics about the benchmark (C.4).

### C.1  Prompt Creation Details

**Step 1: Develop In-Context Examples**   We crafted high-quality in-context (IC) examples with supporting reasoning, which we call 'processes', to guide the prompt generation. These processes are Chain of Thought reasonings that describe the process by which we came up with the prompt. One of the IC Prompts and a corresponding CoT is shown in Appendix C.2

**Step 2: Generate Initial Prompts**   Using the IC examples, we generated synthetic prompts, ensuring diversity by selecting random subsets of IC examples. Each subset used distinct in-context prompts and randomly sampled tools from its set of available tools. The seed prompt used in this step in Appendix C.3.

**Step 3: Filtering**   We manually inspected each prompt to ensure they were reasonable, interesting, and challenging, labeling them as Good, Too Simple, or Nonsensical with justifications for each classification. These labeled examples served as IC inputs for GPT-4 Turbo (OpenAI et al., 2024) to classify additional prompts. We iteratively review the outputs, make necessary edits, and add more IC examples. Through three iterations, the filtered prompts were of high quality, exhibiting only minor mistakes.

**Step 4: Human Refinement**   After filtering, annotators reviewed the finals prompts to resolve any issues related to complexity, clarity and ambiguity. We gave clear instructions on ambiguity (only one possible correct answer) and complexity (requires two or more tool calls to answer), instructing our annotators to ensure the prompt has only one correct answer that is complex, challenging and requires the use of tools.

### C.2  In Context Example

**Prompt**

> I wanna know if eating meat is correlated with heart issues, find the annual per capita consumption of meat in (kg/person) and also the per capita heart attack rates (in heart attacks/person) for every country. Then run a linear regression with y as heart attack rates and x as meat consumption, return the Pearson's correlation as well as the slope of the fit line.

**Process**

> I will first start by creating a prompt that requires the use of google search. I want to make this prompt about investigating whether the amount of meat you consume is correlated to heart disease. In order to make sure there is only one possible answer, I will ask to find the per capita consumption of meat (in kg/person) and heart attacks rates (heart attacks per person) in all countries. This standardizes the actual data that needs to be pulled and specifies the units to ensure there is only one possible answer. I will then ask for a linear regression using that data since it requires a python interpreter. Since linear regression is deterministic when the data is fixed and the data required to fit the linear regression is well defined, I can ask to output itś parameters and ensure there is only one possible answer that can be returned. This ensures that the good prompt is clear, unambiguous and has an answer that is easy to verify through an exact string match while also requiring a chain of dependent tool calls (google search call, then python interpreter call) to solve.

## C.3 SEED PROMPT

> I want you to act as a Prompt Writer.
>
> Please adhere to the following instructions:
>
> - Write a prompt that requires the use of all of the tools.
> - The prompt should require a chain of dependent tools calls who's outputs influence the inputs of the next tool invocation.
> - The prompt should be appropriate for someone in {grade}.
> - Please do not specify the tools to be used in the prompt. We want the assistant to figure out on it's own what tools to call so it should not be specified in the prompt itself. No phrases like "Use the ... tool" should be in the written prompt.
> - The prompt should be a couple sentences.
> - Make sure the prompt has only one possible answer that is concrete and easily verifiable. We want to be able to check the final answer using exact match.
> - Make sure the answer is not in the prompt.
> - Place [STOP] at the end of the prompt.
>
> Examples:
>
> {examples}
>
> [BEGIN ALLOWED TOOLS]
>
> {tools}
>
> [END ALLOWED TOOLS]

## C.4 BENCHMARK METADATA

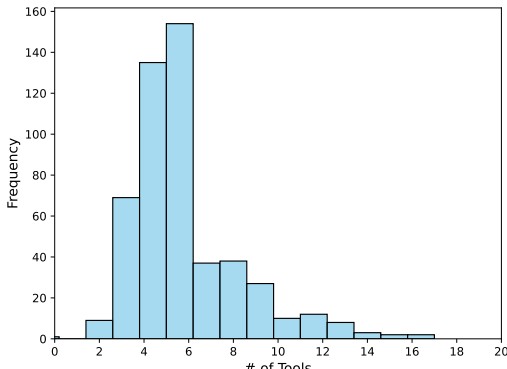

Figure 11: About 85% of prompts in ToolComp require at least 3 tool calls to solve, indicating that they have a decent amount of complexity and difficulty. Furthermore, 20% of prompts still require 7 or more tool calls to solve. This indicates that an agent being evaluated on this benchmark requires high context length, sophisticated reasoning over long context, and advanced tool calling capabilities in order to process long tool chains, formulate a high level plan, and understand the outputs of each tool call to proceed to the next step and subsequently achieve a high score.

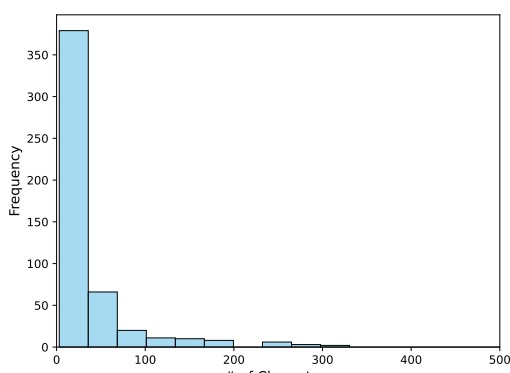

Figure 12: Due to the nature of ToolComp needing to have answers that are easily verifiable, we choose to create prompts that have numbers and short strings to match. However, there are still some examples of prompts that require long structured outputs such as dictionaries, tuples and lists. These test the agent's ability to follow complex queries that involve returning long outputs such as lists or dictionaries of city names, temperatures, altitudes, etc.

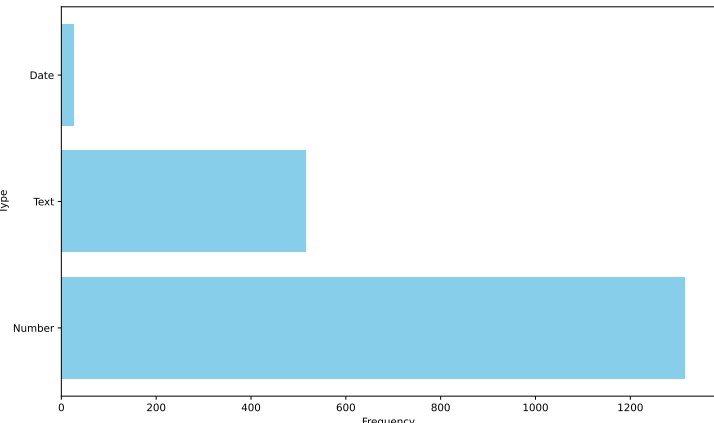

Figure 13: We show the distribution of the following primitive data types: number, string and date. We care most about evaluation of compositional tool use and reasoning rather than aesthetic output structuring and formatting. This is why the benchmark's labels are predominantly numeric while containing a significant fraction of string outputs. In many cases, strings and names are intermediary outputs, but we most often ask for numerical final answers to make the answer easier to unambiguously verify.

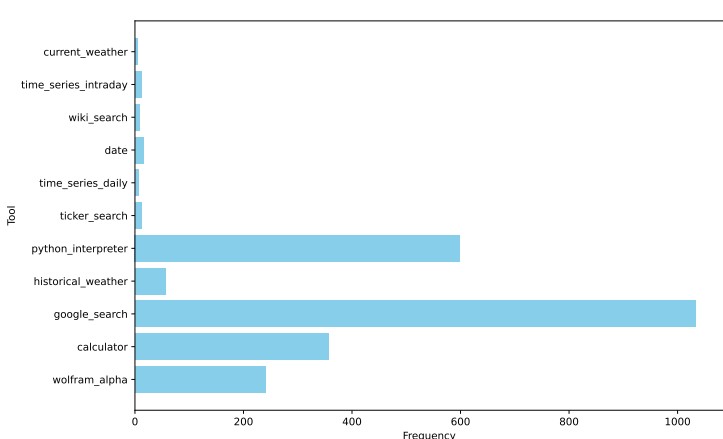

Figure 14: The distribution of tools called in our human supervised tool call chains. The heavy bias towards Google and Python are due to ToolComp Chat only allowing these tools as well them being generally applicable for a wide range of tasks (web retrieval and information processing).

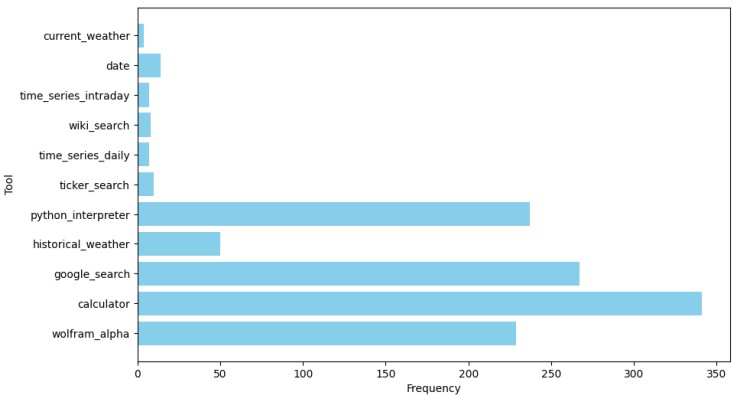

Figure 15: The distribution of tools called in our human supervised tool call chains for just the ToolComp Enterprise subset.

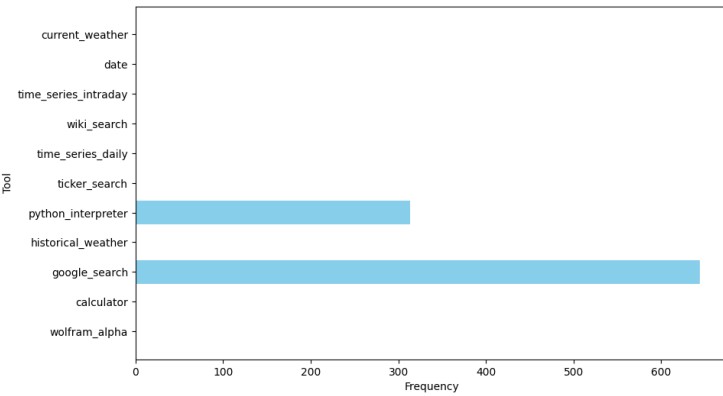

Figure 16: The distribution of tools called in our human supervised tool call chains for just the ToolComp Chat subset.

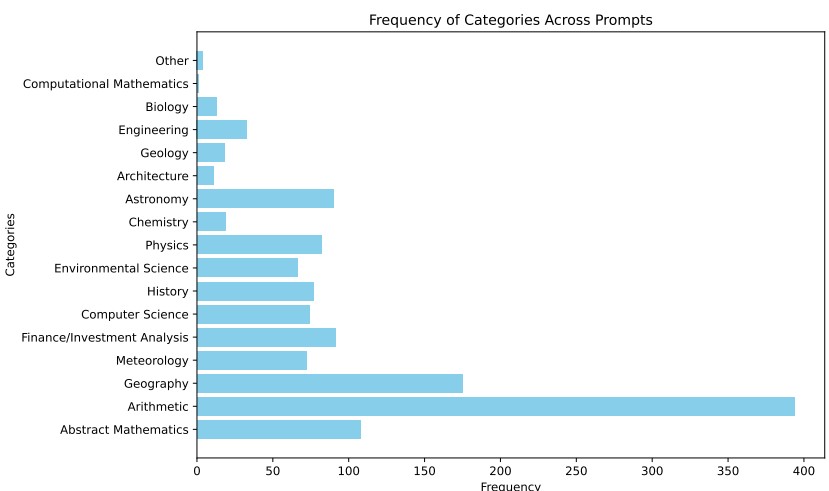

Figure 17: Here, we show the various topics our prompts address. Many prompts require arithmetic operations and mathematical reasoning along with a somewhat uniform distribution of multiple disciplines ranging from Geography, Finance, History, Physics, Chemistry, Astronomy, Architecture etc. The topics are not mutually exclusive since many of these prompts span multiple domains and require multiple tools, multiple sources of knowledge and diverse forms of reasoning.

## D    PROCESS SUPERVISION VS. OUTCOME SUPERVISION TRAINING DETAILS

### D.1    DETAILED SYNTHETIC TRAINING DATA GENERATION

**Synthetic Prompt Generation**    For the generation of synthetic prompts, we mirror the strategies outlined in Section 3.3 with the notable exception of the Human Refinement step due to the high level of associated cost. Instead, we replace this step with final answer consistency across different model families. Using the following models – GPT-4o (May 2024), GPT- 4 Turbo, Claude 3.5 Sonnet, and Llama 3.1 70b – we generate full trajectories and only keep the prompts for which every model arrives at the same final answer. From empirical evaluation, this serves as a good proxy for unambiguous and sensible prompts. Table 8 notes the initial amount of prompts generated and the final number of prompts that are final answer consistent across the model families.

Table 8: Count of training data through the different stages of generation.

| Type | Count |
| --- | --- |
| Initial Prompts | 75K |
| Final Answer Consistent Prompts | 17369 |
| Trajectories with Final Answers | 13628 |
| Trajectories with Correctly Formatted Final Answers | 11654 |

**Synthetic Training Data**    Suppose we have a LLM that acts as a policy, coined the Policy Model, and another LLM that acts as a judge, coined the Critic Model. Given a query, $q$, we first use the Policy Model to generate an action plan, $a$. Then, conditioned on the action plan, we prompt the Policy Model to generate a full chain trajectory $\{t_1, \ldots, t_N\}$, where each $t_i$ is ReAct step that invokes a tool call or invokes the finish action. The prompt to generate the action plan and each tool call is given in E.1 and E.2, respectively. We bound $N$ by 15, allowing at most 15 tool call in trajectory. If the model reaches a final answer, then $t_N$ is the finish action. We then use the Critic Model to critique the action plan $a$ and react steps $t_i$. The prompt for the Critic Model to critique the action plan and the react steps is given in E.3 and E.4, respectively. In the case that the critique model finds a fault with a step, it then proposes a corrected step. The corrected step is then used to continue the chain. Using the latest correct step, the Policy Model will then be invoked to generate

a new ReAct step and the Critic Model will critique the step iteratively until either the Policy Model reaches a final answer and the Critic Model agrees or the Critic Model proposes a final answer when correcting a step. All Policy Model full trajectories that did not reach a final answer was discarded and all Critic Model trajectories that did not reach a final answer was also discarded. For the Policy Model, we used Llama-3.1-8b-Instruct, and for the Critic Model, we used GPT-4o. The process described here mimics the same process we used to create ToolComp described in Figure 1, where human annotators is replaced by GPT-4o and the policy is Llama-3.1-8b-Instruct. Table 9 shows the different generation parameters used for the Policy Model and the Critic Model. From Table 8, the total number of trajectories with correctly formatted final answers is 10342.

Table 9: Policy Model and Critic Model Generation Parameters

| Parameter | Policy Model | Critic Model |
|---|---|---|
| Temperature | 0.5 | 0 |
| Max New Tokens | 1024 | 4096 |
| Num Retries Per Step | 3 | 3 |
| Stop | "End Action", "End Action\n", "\nEnd Action" | "End Action" |

### D.2 Construction of Training Datasets

**Reward Model Datasets**   The preference ranking dataset for outcome reward modelling (ORM) is composed of the dis-preferred trajectory being the original Policy Model full trajectory and preferred trajectory being the fully corrected Critic Model trajectory (assuming at least one edit was made in the duration of the Critic Model trajectory). The step-by-step preference ranking dataset for the process reward modelling (PRM) is comprised of every Critic Model correction, we take the dis-preferred step to be the original Policy Model's full ReAct step and the preferred step to be the Critic Model's full ReAct corrected step. The trajectory history is the most correct chain leading up to the given step. In total, as in Table 10, the ORM dataset size is same as the total number of trajectories with correctly formatted final answers (11654) and the PRM dataset size is equivalant to the number of times the Policy and Critic models had disputes and corrections (14932). We note that the ORM samples are full trajectories, whereas the PRM samples are simply the steps that were correct.

**SFT Dataset**   The SFT dataset is comprised of the most corrected trajectory taken by simply following all Critic Model correct steps where applicable.

Table 10: Training dataset sizes.

| Dataset | Number of Samples |
|---|---|
| ORM | 11654 |
| PRM | 14932 |
| SFT | 11654 |

### D.3 Detailed Training Objectives

**Outcome Reward Model**   The base model is equipped with a linear layer to serve as the reward head, which outputs a scalar reward value per token. We place a special RM token at the end of the preferred and dis-preferred completion, followed by an EOS token. The training objective is then given by a Binary Cross Entropy loss on the reward value output on the special RM token. We use Llama-3.1-8b-Instruct as the base model.

**Process Reward Model**   Similarly, the base model is equipped with a linear layer to serve as the reward head, which outputs a singular scalar value. We test 4 different levels of supervision. For the

Full Step with Observation and Full Step without Observation, a singular special RM token is placed at the end of the observation step and at the end of the action input step, respectively. Moreover, for Sub Step with Observation and Sub Step without Observation, 4 special RM tokens are placed after each ReAct step (including the observation) and 3 special RM tokens are placed after each ReAct step (excluding the observation), respectively. For all variation, the training objective is given by the average Binary Cross Entropy loss across the reward value output at each of the special RM tokens and the corresponding label. We use Llama-3.1-8b-Instruct as the base model.

## D.4 PRM vs. ORM Evaluation

**Ranking Trajectories**    We use a generator to generate 50 completions per problem in ToolComp. Then for each completion, we discard all completions that did not reach a final answer. We then use an ORM and PRM to rank the completions per problem. For the ORM, we simply take the sigmoid of the reward score it places on the completions as that is the probability the model assigns that the trajectory is correct. For PRM, given a trajectory, we collect a list of probability scores for each step. To combine the list of probability scores into a single number, we experiment with a couple of aggregation functions, namely: min, max, average, and product. In order to account for variance in the ranking scores, we perform 500 permutations of the completion per problem, leaving 20 out at random each permutation. We consider the best-of-30 accuracy, which takes the best trajectory as ranked by the corresponding method and evaluates the correctness of that single trajectory.

**Training Dataset Scales**    In order to assess the generalizability of PRM vs. ORM with increasing scales of data, we vary the dataset sizes to be 10, 25, 50, and 100 percent of the full dataset. At each dataset scale, we train the models for 3 epochs and then take the best performing model on a held out synthetically generated validation set.

**Base Model vs. SFT Model Generations**    To account for a variety in the quality of Tool-Use trajectories to rank and to assess the performance gains beyond SFT, in addition to using Llama-3.1-8b-Instruct model as a generator, we also use a Supervise Fine-Tuned Llama-3.1-8b-Instruct. The model is trained on the full set of golden trajectories given by following all the Critic Model's corrections.

## D.5 Training Implementation and Hyper-parameters

**Implementation**    We implement the reward model and SFT training using the OpenRLHF library (Hu et al., 2024) in combination with PyTorch (Paszke et al., 2019). We made the following modification to the OpenRLHF Library:

- The RewardDataset object optionally took indices that signified where to place the special RM tokens to implement the ORM training and the 4 different PRM supervision training

- We update the BCE loss function implementation to additional consider more than one RM classification token id for the PRM training. We simply average the BCE loss for the multiple settings.

**Training Parameters and Selection**    We use Llama-3.1-8b-Instruct weights as initialization for both the ORM and the PRM. For each of the ORM and PRM training across the different dataset scales, we keep a constant batch size of 64 with 3 epochs and experiment with 5 different learning rates: $1e{-}5$, $1e{-}6$, $9e{-}7$, $6e{-}7$, and $1e{-}7$. We then pick the best respective model using the best performance on a held-out (10%) validation set. Generally speaking, the best learning rate for PRM training is $9e{-}7$ and for ORM training is $1e{-}6$ across dataset scales. For the SFT training we experiment with 3 different learning rates $1e{-}5$, $1e{-}6$, and $1e{-}7$ and 5 epochs. Table 11 and Table 12 summarizes hyper-parameters and training settings.

Table 11: Training settings for ORM and PRM.

| hyper-parameter and settings | value |
|---|---|
| epochs | 3 |
| context length | 16384 |
| batch size | 64 |
| deepspeed zero stage | 2 |
| seed | 42 |
| loss function | BCE |
| learning rates | 1e−5, 1e−6, 9e−7, 6e−7, 1e−7 |

Table 12: Training settings for SFT.

| hyper-parameter and settings | value |
|---|---|
| epochs | 5 |
| context length | 16384 |
| batch size | 64 |
| deepspeed zero stage | 3 |
| seed | 42 |
| loss function | CE |
| learning rates | 1e−5, 1e−6, 1e−7 |

# E  TOOL-USE PROMPTS

In this section, we summarize all of the prompts that were used during the creation of the benchmark, evaluation of the benchmark, and creation of the synthetic training data. For the creation of the benchmark, we state the "Action Plan Prompt" for the Policy Model in Section E.1 and the "Tool Call Prompt" for the Policy Model in Section E.2. For the evaluation of the benchmark, we state the LLM grading prompt and the in-context examples used to aid grading in Section E.5. Lastly, for the creation of the synthetic training data, we use the same policy model prompts for the action plan and tool call, and we additionally include the "Action Plan Prompt" for the Critic Model in Section E.3 and the "Tool Call Prompt" for the Critic Model in Section E.4.

## E.1  ACTION PLAN PROMPT (POLICY MODEL)

You are a helpful action planner with access to functions. Please use the tools to provide information accurate up to current date: {current_date}

FUNCTIONS: {func_spec}

Question: {question}

Given the tools available to you above, please formulate an action plan to answer the question in a bulleted list for each step. Refrain from using any specific tool calls in your action plan, instead focus on the high-level steps you would take to answer the question and the name of the tool you would use and how you would use it. Refrain from trying to answer the question directly in the action plan.

## E.2 REACT TOOL CALL PROMPT (POLICY MODEL)

SYSTEM:

You are a helpful assistant with access to functions, each function will be regarded as an action. Your job is to take relevant and necessary actions to get to the final answer to a user question. Please use the actions to provide information accurate up to current date and time: {current_date}. The user will provide you a question and a high level action plan. Your job is to execute on the action plan to answer the question. It's okay to slightly deviate from the action plan if you think it's necessary.

FUNCTIONS: {func_spec}

Please stick to the following format:

Thought: ⟨ your reasoning/thought on why/how to use an action⟩
Action: ⟨the action to take, should be one of {func_list}⟩
Action Input: ⟨the input to the action (should be in JSON format with the required fields)⟩
End Action

If you believe that you have obtained enough information (which can be judged from the history observations) to answer the question, please call:

Thought: I have enough information to answer the question
Action: finish
Action Input: {"answer": [your answer string]}}
End Action

For your final answer (the finish action input), make sure you answer the full question. Additionally, we want to make sure the final answers/outputs in the finish action input are returned in the order that they are given in a list format so we can verify them with an exact string match. For eg. if the prompt asks for a city name, its temperature and a list of names of all the NBA teams whose home stadium is within a 400 mile radius, you would output ['San Francisco', 78, ['Los Angeles Lakers', 'Golden State Warriors']].

If the prompt asks for a special sorting of the list, make sure to output wrap the list in {{}} and if doesn't require any special sorting wrap it in [] like you normally would. So if the prompt instead asked to list the names of all the NBA teams whose home stadium is within a 400 mile radius in alphabetical order, you would output [San Francisco, 78, {{Golden State Warriors, Los Angeles Lakers}}].

Only output the final answer with no additional text or natural language. Give dates in YYYY-MM-DD format, temperatures in celcius, prices in dollars, lengths in meters, area in meters$^2$, volume in $m^3$ and angles in degrees if the prompt doesn't specify what format/units to output the answer in.
Given a user provided question and action plan, as well as your previous actions and observations, take your next action.

USER:

Question: {question}

Action Plan: {action_plan}

ASSISTANT:

{history_of_react_steps}

### E.3 ACTION PLAN PROMPT (CRITIC MODEL)

You are an expert planner of tool calls. Your job is to critique the action plan of an assistant.

The following information is shown to the assistant in order to devise an action plan:

[Start of the message]

You are a helpful assistant with access to functions. Please use the tools to provide information accurate up to current date and time: {current_date}.

FUNCTIONS: {func_spec}

Question: {question}

Given the question and the tools available to you above, please formulate an action plan to answer the question in a bulleted list for each step.

Refrain from using any specific tool calls in your action plan, instead focus on the high-level steps you would take to answer the question and the name of the tool you would use and how you would use it. Refrain from trying to answer the question directly in the action plan.

[End of the message]

Given the set of functions and the question, please critique the action plan provided by the assistant.

First, determine if the action plan is correct or incorrect. To do so, provide a reasoning and then label the action plan as correct or incorrect. In order to determine if the action plan needs revision, consider the following:

- Is the action plan reasonable given the set of functions available?
- Is the action plan clear and concise?
- Is the action plan missing any steps?

Please err on the side of giving the assistant the benefit of the doubt, and only critique the action plan if it is clearly incorrect.
If the action plan is incorrect, provide a revised action plan that you believe would be correct.

Furthermore, your output should follow the format:

Reasoning: ⟨ your reasoning for the correctness or incorrectness of the action plan ⟩

Label: ⟨ correct/incorrect ⟩

Revised Action Plan: ⟨ your revised action plan or empty if no revision needed ⟩

Here is the action plan provided by the assistant:

{action_plan}

Please provide your critique of the action plan.

### E.4  REACT TOOL CALL PROMPT (CRITIC MODEL)

You are an expert judge of tool calls. Your job is to critique each of the ReAct steps of an assistant.

The following information is shown to the assistant in order to devise a ReAct step.

[Start of the message]

You are a helpful assistant with access to functions. Use them if required. Please use the tools to provide information accurate up to current date and time: {current_date}.

FUNCTIONS: {func_spec}

Please stick to the following format:

Thought: you should always think about what to do
Action: the action to take, should be one of {func_list}
Action Input: the input to the action
End Action

If you believe that you have obtained enough information (which can be judged from the history observations) to answer the question, please call:

Thought: I have enough information to answer the question
Action: finish
Action Input: "answer": [your answer string]
End Action

Question: {question}

[End of the message]

Given the set of functions, question, action plan and history of past actions, critique the Thought, Action, and Action Input step. Assume the action plan and history of past actions are optimal. To assess the thought step, if the step is roughly reasonable and the action and action input step are correlated with the thought step, then the thought step is correct. Please give the assistant the benefit of the doubt and be lenient in your assessment.

To assess the action step, let's assume that the Assistant cannot complete simple functionalities such as simple arithmetic, converting units, or utilizing simple facts without the use of tools. If the action specifies a reasonable function to use, then the action step is correct.

To assess the action input step, if the input is reasonable and the action is correct, then the action input step is correct.

If any of the steps are incorrect, label them as incorrect in the Labels section.

For the Revised ReAct Step section, provide the correct step that the assistant should have taken. If the assistant's step is correct, provide the assistant's step as the revised step. If the assistant's step is incorrect, provide the correct step that the assistant should have taken. As a general rule of thumb, if your revised step is different from the assistant's step, then the assistant's step is incorrect, and if your revised step is the same as the assistant's step, then the assistant's step is correct.

As an important reminder, for your final answer (the finish action input), we want to make sure the final answers/outputs in the finish action input are returned in the order that they

are given in a list format so we can verify them with an exact string match. For eg. if the prompt asks for a city name, its temperature and a list of names of all the NBA teams whose home stadium is within a 400 mile radius, you would output ['San Francisco', 78, ['Los Angeles Lakers', 'Golden State Warriors']]. If the prompt asks for a special sorting of the list, make sure to output wrap the list in {{}} and if doesn't require any special sorting wrap it in [] like you normally would. So if the prompt instead asked to list the names of all the NBA teams whose home stadium is within a 400 mile radius in alphabetical order, you would output [San Francisco, 78, {{Golden State Warriors, Los Angeles Lakers}}].

Only output the final answer with no additional text or natural language or units. Give dates in YYYY-MM-DD format, temperatures in Celcius, prices in dollars, lengths in meters, area in meters$^2$, volume in $m^3$ and angles in degrees if the prompt doesn't specify what format/units to output the answer in.

As a reminder, you should not use an external information that is not provided in the prompt or by a tool call. As a simple example, you may know a ticker symbol already for a company, but you should not use it unless you have called the ticker_search or a similar function (e.g. google_search, wiki_search, etc.) to retrieve that information.

Your output should follow the format:

[Start of format]

Reasoning: ⟨ your reasoning for the correctness or incorrectness of each step ⟩

Labels: [⟨correct/incorrect⟩, ⟨correct/incorrect⟩, ⟨correct/incorrect⟩] (in the order of Thought, Action, Action Input)

Revised ReAct Step:

Thought: ⟨ your revised thought or assistant's thought if correct ⟩
Action: ⟨ your revised action or assistant's action if correct ⟩
Action Input: ⟨ your revised action input or assistant's action input if correct ⟩
End Action

[End of format]

Here is the action plan:

{action_plan}

Here is the history of past actions. If there are no past actions yet, this will be empty:

{history}

Here is the latest ReAct step provided by the assistant:

Thought: {thought}
Action: {action}
Action Input: {action_input}
End Action
Observation: {observation}

Please provide your critique of the latest ReAct step provided by the assistant.

## E.5 LLM GRADING PROMPT

### E.5.1 MAIN PROMPT

You are an expert test grader. You have been given a student answer ('Student Answer:') to grade. You have also been the correct answer ('Correct Answer:') and the original question ('Question:'). Each correct answer is a list of strings.

**{In-Context Examples}**

The possible grades are

**INCORRECT**: 'Student Answer:' is different from 'Correct Answer:'

- numbers are completely different
- lists are completely different
- 'Question:' asks for special sorting of a list but the list in 'Student Answer:' is sorted differently than 'Correct Answer:'
- strings are completely different or information present in the string is completely different

**CORRECT BUT BAD FORMATTING**: 'Student Answer:' has the same info as 'Correct Answer:' but is formatted differently.

- 'Student Answer:' includes natural language or additional text
- numbers are formatted differently but they are close to one another ('Student Answer:' is within
- lists are wrapped differently than the correct answer but contains the same information and sorted the same way as 'Correct Answer:' if asked 'Question:' asks for a special sorting
- Strings are the same but may be formatted differently

**CORRECT**: The student answer has the same info as 'Correct Answer:' and is also formatted the same as 'Correct Answer:'

- numbers are close to one another ('Student Answer:' is within 10% of the correct answer)
- if 'Question:' asks for a special sorting of the list the 'Student Answer:' list is sort the same as 'Correct Answer:'
- lists are wrapped the same
- Strings are identical

Remember you are assuming the correct answer provided is correct, your job is is only to compare the correct answer to the student answer and grade it based on the above criteria. Do not try to determine the correct answer yourself. Make sure to include a reasoning and final grade in the format:

Reasoning: ⟨ reasoning ⟩ Final Grade: ⟨ INCORRECT / CORRECT BUT BAD FORMATTING / CORRECT ⟩ [ENDOFGRADE]

Now do this for the following user provided question, student answer and correct answer.

### E.5.2 IN-CONTEXT EXAMPLES (ORDERING)

We want to make sure the values in the student answer are returned in the order that they are asked in 'Question:'.

For example, if 'Question:' asks for a city name, its temperature and a list of names of all the NBA teams whose home stadium is within a 400 mile radius, and 'Correct Answer:' is ['San Francisco', 78, ['Golden State Warriors', 'Los Angeles Lakers']] we would want 'Student Answer:' can be ['San Francisco', 78, ['Los Angeles Lakers', 'Golden State Warriors']].

Examples:

**Question**: Find the name of the city known for its famous tourist attraction Alcatraz, also give it's current temperature and a list of names of all the NBA teams whose home stadium is within a 400 mile radius
**Correct Answer**: ['San Francisco', 78, ['Golden State Warriors', 'Los Angeles Lakers']
**Student Answer**: ['San Francisco', 74, ['Los Angeles Lakers', 'Golden State Warriors']]
**Reasoning**: The Student Answer is correct because it identifies the same city , the temperature is within 10% of the Correct Answer and the same team names are present in the list.
**Final Grade**: CORRECT

**Question**: Find the name of the city known for its famous tourist attraction Alcatraz, also give it's current temperature and a list of names of all the NBA teams whose home stadium is within a 400 mile radius
**Correct Answer**: ['San Francisco', 78, ['Golden State Warriors', 'Los Angeles Lakers']
**Student Answer**: The city name is San Francisco, its temperature is 80 degrees and the Los Angeles Lakers and the Golden State Warriors are two NBA teams whose home stadium is within a 400 mile radius
**Reasoning**: Although the Student Answer is correct (identifies the same city , the temperature is within 10% of the Correct Answer and the same team names are present), it's not formatted the same and contains extra text and natural language.
**Final Grade**: CORRECT BUT BAD FORMATTING

**Question**: Find the name of the city known for its famous tourist attraction Alcatraz, also give it's current temperature and a list of names of all the NBA teams whose home stadium is within a 400 mile radius
**Correct Answer**: ['San Francisco', 78, ['Golden State Warriors', 'Los Angeles Lakers']
**Student Answer**: ['San Francisco', -15, ['Los Angeles Lakers', 'Golden State Warriors']]
**Reasoning**: The Student Answer is incorrect because although identifies the same city and the same team names are present in the list, the temperature is well outside of 10% of the Correct Answer.
**Final Grade**: INCORRECT

### E.5.3 IN-CONTEXT EXAMPLES (SORTING)

If 'Question:' asks for a special sorting of the list, make sure 'Student Answer:' is sorted the same as 'Correct Answer:'. So if 'Question:' instead asked to list the names of all the NBA teams whose home stadium is within a 400 mile radius in alphabetical order, we would want 'Student Answer:' to contain ['San Francisco', 78, ['Golden State Warriors', 'Los Angeles Lakers']].

Examples:

**Question**: Find the name of the city known for its famous tourist attraction Alcatraz, also give it's current temperature and a list of names of all the NBA teams whose home stadium is within a 400 mile radius in alphabetical order
**Correct Answer**: ['San Francisco', 78, ['Golden State Warriors', 'Los Angeles Lakers']
**Student Answer**: ['SF', 75, ['Golden State Warriors', 'Los Angeles Lakers']]
**Reasoning**: The Student Answer is correct because it identifies the same city (SF is a commonly known short form for San Francisco), the temperature is within 10% of the Correct Answer and the same team names are present in the list and sorted the same given the Question asks for a special sorting (alphabetical order).
**Final Grade**: CORRECT

**Question**: Find the name of the city known for its famous tourist attraction Alcatraz, also give it's current temperature and a list of names of all the NBA teams whose home stadium is within a 400 mile radius in alphabetical order
**Correct Answer**: ['San Francisco', 78, ['Golden State Warriors', 'Los Angeles Lakers']
**Student Answer**: The city name is San Francisco, its temperature is 80 degrees and the Golden State Warriors and the Los Angeles Lakers are two NBA teams whose home stadium is within a 400 mile radius (in alphabetical order)
**Reasoning**: Although the Student Answer is correct (identifies the same city , the temperature is within 10% of the Correct Answer and the same team names are present in the same order), it's not formatted the same and contains extra text and natural language.
**Final Grade**: CORRECT BUT BAD FORMATTING

**Question**: Find the name of the city known for its famous tourist attraction Alcatraz, also give it's current temperature and a list of names of all the NBA teams whose home stadium is within a 400 mile radius in alphabetical order
**Correct Answer**: ['San Francisco', 78, ['Golden State Warriors', 'Los Angeles Lakers']
**Student Answer**: ['San Francisco', 79, ['Los Angeles Lakers', 'Golden State Warriors']]
**Reasoning**: The Student Answer is incorrect because although it identifies the same city and the temperature is within of 10% of the Correct Answer, the list of teams name is in a differently ordered even though 'Question:' explicitly asks for a special sorting.
**Final Grade**: INCORRECT

# F  TOOLCOMP EXAMPLES

## F.1  EXAMPLE GOLDEN TRAJECTORY

**Question:** Suppose you have a $2\,\text{inch} \times 2\,\text{inch} \times 2\,\text{inch}$ box. How many of these boxes can you fit into the biggest U-Haul truck (biggest truck as of 2024)?

**Action Plan**

1. Use the `google_search` tool to find the name of the biggest U-Haul truck and find its dimensions.
2. Use `python_interpreter` to calculate the total number of boxes that would fit given the dimensions.
3. Return the answer.

**ReAct Step 1**

- **Thought:** I need to find the dimensions of the biggest U-Haul truck.
- **Action:** `google_search`
- **Action Input:** {"query": "biggest U-Haul truck 2024"}
- **Observation:** {(search results omitted for brevity)}

**ReAct Step 2**

- **Thought:** I now have the dimensions of the biggest U-Haul truck, so I can use Python to calculate how many boxes can fit inside.
- **Action:** `python_interpreter`
- **Action Input:**

```python
import math
box_length = 2
box_width = 2
box_height = 2
# convert truck dimensions into inches
truck_length = (26 * 12) + 5
truck_width = (7 * 12) + 8
truck_height = (8 * 12) + 3
num_boxes = math.floor(truck_length // box_length) * \
            math.floor(truck_width // box_width) * \
            math.floor(truck_height // box_height)
print(f'Total number of boxes: {num_boxes}')
```

- **Observation:** {"result": "Total number of boxes: 356132", "error": null}

**ReAct Step 3**

- **Thought:** I have the total number of boxes that can fit inside the truck, so I should return that as the answer.
- **Action:** `finish`
- **Action Input:** {"answer": "356132"}
- **Observation:** 356132

## F.2 EXAMPLE ANNOTATED TRAJECTORY

> **Question:** Using the 2023 birth-rate, death-rate, and population of Japan, assuming no one leaves or enters, what will be the projected population in 10 years?

**Action Plan**

| Model |
| --- |
| 1. Use the `python_interpreter` tool to calculate the population growth rate using the birth-rate, death-rate, and population of Japan.

2. Use the `python_interpreter` tool to calculate the population in 10 years using the population growth rate and the current population of Japan.

3. Use the `google_search` tool to search for the current birth-rate, death-rate, and population of Japan.

4. ... (truncated since the model goes in circles)

(**Label**: ✗) |

| Human |
| --- |
| 1. Use the `google_search` tool to calculate the population growth rate using the birth-rate, death-rate, and population of Japan in 2023.

2. Use the `python_interpreter` tool to calculate the population in 10 years using the population growth rate and the population of Japan in 2023.

3. Use the finish action to return the final answer. |

**ReAct Step 1**

| Model |
| --- |
| • **Thought**: I need to use the `google_search` tool to calculate the population growth rate using the birth-rate, death-rate, and population of Japan in 2023. (**Label**: ✓)

• **Action**: `google_search` (**Label**: ✓)

• **Action Input**: { "query": "Japan 2023 population growth rate using birth-rate, death-rate, and population" } (**Label**: ✗)

• **Observation**:
`[ {"source": "nippon.com", "date": "Mar 13, 2024", "title": "Number of Births in Japan Reaches New Low in 2023", "snippet": "In 2023, Japan had more than twice as many deaths as births, leading to a natural population decline (excluding migration) of over 800000.", "highlight": "had more than twice as many deaths as births"}, ... truncated]` |

| Human |
| --- |
| • **Thought**: N/A

• **Action**: N/A

• **Action Input**: { "query": "Japan 2023 birth-rate per 1000" }

• **Observation**:
`[{ "source": "statista.com", "date": "Jul 04, 2024", "title": "Japan: total population 2019-2029", "snippet": "In 2023, the total population of Japan amounted to around 124.62 million inhabitants. See the figures for the population of South Korea for ...", "highlight": "124.62 million inhabitants" }, ... truncated]` |

**ReAct Step 2**

<table>
<tr><td>

**Model**

- **Thought**: I need to use the `google_search` tool to calculate the population growth rate using the birth-rate, death-rate, and population of Japan in 2023. (**Label**: ✗)

- **Action**: `google_search` (**Label**: ✓)

- **Action Input**: { "query": "Japan 2023 death-rate per 1000" } (**Label**: ✗)

- **Observation**:
```
[ {"source":
"data.worldbank.org",
"date": null, "title":
"Death rate, crude (per 1000
people) - Japan", "snippet":
"Death rate, crude (per
1000 people) - Japan from
The World Bank: Data.",
"highlight": "Death rate
| per 1000 | Japan" }, ...
truncated]
```

</td><td>

**Human**

- **Thought**: According to wikipedia the 2023 estimates for the birthrate 6.6 births/1000 population and 11.7 deaths/1000 population. I need to find the 2023 population for Japan.

- **Action**: N/A

- **Action Input**: { "query": "Japan 2023 population" }

- **Observation**:
```
[{ "source":
"statista.com", "date":
"Jul 04, 2024", "title":
"Japan: total population
2019-2029", "snippet": "In
2023, the total population
of Japan amounted to around
124.62 million inhabitants.
See the figures for the
population of South Korea
for ...", "highlight":
"124.62 million inhabitants"
}, ... truncated]
```

</td></tr>
</table>

**ReAct Step 3**

<table>
<tr><td>

**Model**

- **Thought**: I need to use the python interpreter tool to calculate the population in 10 years using the population growth rate and the population of Japan in 2023. (**Label**: ✓)

- **Action**: `python_interpreter` (**Label**: ✓)

- **Action Input**:
```
import numpy as np
population = 124352000
population_growth_rate =
0.0012
population_in_10_years
= population * (1 +
population_growth_rate)**10
print(population_in_10_years)
```
(**Label**: ✗)

- **Observation**: | 125852307.84945872 |

</td><td>

**Human**

- **Thought**: N/A

- **Action**: N/A

- **Action Input**:

```
population = 124620000
death_rate_per_thousand =
11.7 total_births_extrapolated
= birth_rate_per_thousand
* (population / 1000)
total_deaths_extrapolated
= death_rate_per_thousand
* (population / 1000)
population_growth_rate =
(total_births_extrapolated
- total_deaths_extrapolated)
/ population
population_in_10_years
= population * (1 +
population_growth_rate)**10
print(population_in_10_years)
```

- **Observation**: | 118408275.35965854 |

</td></tr>
</table>

**ReAct Step 4**

Model

- **Thought**: I have enough information to answer the question (**Label**: ✓)
- **Action**: `finish` (**Label**: ✓)
- **Action Input**: `{"answer": ["118,408,275"]}` (**Label**: ✗)
- **Observation**: $\left[\text{"}118, 408, 275\text{"}\right]$

Human

- **Thought**: N/A
- **Action**: N/A
- **Action Input**: `{"answer": ["118408275"]}`
- **Observation**: $\left[\text{"}118408275\text{"}\right]$

# G TOOLS

In this appendix section, we provide a detail breakdown of each of the 11 tools, providing a description, the parameters, an input example and a corresponding output example.

## G.1 DATE

**Description:** Returns the current date (e.g., January 1, 2024).

**Input Example:**

```
{}
```

**Output Example:**

```
{
  "result": "Today is Friday, February 16, 2024.",
  "error": ""
}
```

**Parameters:**

```
[]
```

## G.2 CALCULATOR

**Description:** Calculates expressions including basic arithmetic and brackets.

**Input Example:**

```
{
  "operation": "2*32-4+456+(1+2)+3+(1/2*3+3+(1+2))"
}
```

**Output Example:**

```
{
  "error": "",
  "result": "529.5"
}
```

**Parameters:**

```
[
  {
    "Input Name": "operation",
    "Type": "String",
    "Description": "Computes numerical expressions involving float
        numbers and operators like +, -, *, /, ^.\""
  }
]
```

### G.3 CURRENT WEATHER

**Description:** Retrieves current daily averages for temperature, rainfall, and hours of precipitation for a specified city and country. Does not return historical data.

**Input Example:**

```
1  {
2    "city_name": "London",
3    "country_code": "GB"
4  }
```

**Output Example:**

```
1  {
2    "error": "",
3    "result": [
4      {
5        "date": "2024-03-25 00:00:00",
6        "temperature (F)": "47.78615",
7        "total rain (mm)": "1.4000001",
8        "total snowfall (mm)": "0.0",
9        "precipitation hours (hours)": "4.0"
10     },
11     {
12       "date": "2024-03-26 00:00:00",
13       "temperature (F)": "48.374897",
14       "total rain (mm)": "8.2",
15       "total snowfall (mm)": "0.0",
16       "precipitation hours (hours)": "11.0"
17     },
18     {
19       "date": "2024-03-27 00:00:00",
20       "temperature (F)": "47.217274",
21       "total rain (mm)": "2.399999",
22       "total snowfall (mm)": "0.0",
23       "precipitation hours (hours)": "4.0"
24     }
25   ]
26 }
```

**Parameters:**

```
1  [
2    {
3      "Input Name": "city_name",
4      "Type": "String",
5      "Description": "The name of the city."
6    },
7    {
8      "Input Name": "country_code",
9      "Type": "Two Alphabet-Number",
10     "Description": "The country code (ISO 3166-2). The list can be found
           here: https://en.wikipedia.org/wiki/ISO_3166-2"
11   }
12 ]
```

## G.4 HISTORICAL WEATHER

**Description:** Retrieves daily averages for temperature and precipitation starting from the 1940s for a given city. Note: 5-day data delay, meaning you cannot get current weather data for the last 5 days.

**Input Example:**

```
{
  "city_name": "London",
  "country_code": "GB",
  "start_date": "2023-03-09",
  "end_date": "2023-03-21"
}
```

**Output Example:**

```
{
  "error": "",
  "result": [
    {
      "date": "2024-03-09 00:00:00",
      "temperature (F)": "48.102356",
      "total rain (mm)": "0.4",
      "total snowfall (mm)": "0.0",
      "precipitation hours (hours)": "2.0"
    },
    ...
    {
      "date": "2024-03-23 00:00:00",
      "temperature (F)": "43.373596",
      "total rain (mm)": "1.0999999",
      "total snowfall (mm)": "0.42000002",
      "precipitation hours (hours)": "3.0"
    }
  ]
}
```

**Parameters:**

```
[
  {
    "Input Name": "city_name",
    "Type": "String",
    "Description": "The name of the city."
  },
  {
    "Input Name": "country_code",
    "Type": "Two Alphabet-Number",
    "Description": "The country code (ISO 3166-2). The list can be found
        here https://en.wikipedia.org/wiki/ISO_3166-2"
  },
  {
    "Input Name": "start_date",
    "Type": "Date Format",
    "Description": "The start date in YYYY-MM-DD format"
  },
  {
    "Input Name": "end_date",
    "Type": "Date Format",
    "Description": "The start date in YYYY-MM-DD format"
  }
]
```

### G.5 WIKI SEARCH

**Description:** Searches Wikipedia and returns a summary of the top pages matching the query.

**Input Example:**

```
{
  "query": "covid-19",
  "num_results": "1"
}
```

**Output Example:**

```
{
  "error": "",
  "result": [
    {
      "title": "COVID-19",
      "summary": "Coronavirus disease 2019 (COVID-19) is a contagious
          disease caused by the coronavirus SARS-CoV-2. The first known
          case was identified in Wuhan, China, in December 2019. Most
          scientists believe the SARS-CoV-2 virus entered into human
          populations through natural zoonosis, similar to the SARS-CoV-1
           and MERS-CoV outbreaks, and consistent with other pandemics in
           human history. Social and environmental factors including
          climate change, natural ecosystem destruction and wildlife
          trade increased the likelihood of such zoonotic spillover. The
          disease quickly spread worldwide, resulting in the COVID-19
          pandemic. The symptoms of COVID-19 are variable but often
          include fever, fatigue, cough, breathing difficulties, loss of
          smell, and loss of taste. Symptoms may begin one to fourteen
          days after exposure to the virus. At least a third of people
          who are infected do not develop noticeable symptoms. Of those
          who develop symptoms noticeable enough to be classified as
          patients, most (81%) develop mild to moderate symptoms (up to
          mild pneumonia), ... truncated"
    }
  ]
}
```

**Parameters:**

```
[
  {
    "Input Name": "query",
    "Type": "String",
    "Description": "The search query."
  },
  {
    "Input Name": "num_results (Optional)",
    "Type": "Integer",
    "Description": "Number of search results to return."
  }
]
```

### G.6 INTRADAY STOCK INFO

**Description:** Provides intraday time series data for specified equities.

**Input Example:**

```
{
  "symbol": "AAPL",
  "interval": "60min"
}
```

**Output Example:**

```
{
  "error": "",
  "result": [
    {
      "timestamp": "2024-07-16 19:00:00",
      "open_market_value": "234.6520",
      "high_market_value": "234.7200",
      "low_market_value": "234.2200",
      "close_market_value": "234.3200",
      "volume": "38722"
    },
    {
      "timestamp": "2024-07-16 18:00:00",
      "open_market_value": "234.6220",
      "high_market_value": "234.7500",
      "low_market_value": "234.5050",
      "close_market_value": "234.7000",
      "volume": "24098"
    },
    ...
    {
      "timestamp": "2024-07-08 16:00:00",
      "open_market_value": "227.8100",
      "high_market_value": "227.8800",
      "low_market_value": "226.0630",
      "close_market_value": "227.6400",
      "volume": "14364524"
    }
  ]
}
```

**Parameters:**

```
[
  {
    "Input Name": "symbol",
    "Type": "String",
    "Description": "The ticker symbol of the equity."
  },
  {
    "Input Name": "interval",
    "Type": "String",
    "Description": "Data point interval (1min, 5min, etc.)."
  },
  {
    "Input Name": "month (optional)",
    "Type": "String",
    "Description": "You can use the month parameter (in YYYY-MM format)
        to query a specific month in history."
  }
]
```

## G.7 DAILY STOCK INFO

**Description:** Returns daily time series data for specified equities.

**Input Example:**

```
{
  "symbol": "AAPL",
  "number_of_days": 5
}
```

**Output Example:**

```
{
  "error": "",
  "result": [
    {
      "timestamp": "2024-07-16",
      "open_market_value": "235.0000",
      "high_market_value": "236.2700",
      "low_market_value": "232.3300",
      "close_market_value": "234.8200",
      "volume": "43234278"
    },
    {
      "timestamp": "2024-07-15",
      "open_market_value": "236.4800",
      "high_market_value": "237.2300",
      "low_market_value": "233.0900",
      "close_market_value": "234.4000",
      "volume": "62631252"
    },
    ...
    {
      "timestamp": "2024-07-10",
      "open_market_value": "229.3000",
      "high_market_value": "233.0800",
      "low_market_value": "229.2500",
      "close_market_value": "232.9800",
      "volume": "62627687"
    }
  ]
}
```

**Parameters:**

```
[
  {
    "Input Name": "symbol",
    "Type": "String",
    "Description": "The ticker symbol of the equity."
  },
  {
    "Input Name": "number_of_days",
    "Type": "Integer",
    "Description": "The number of days before today to return data for."
  }
]
```

## G.8 STOCK SYMBOL SEARCH

**Description:** Searches for stock tickers based on provided keywords.

**Input Example:**

```
1  {
2    "keywords": "tesla"
3  }
```

**Output Example:**

```
1  {
2    "error": "",
3    "result": [
4      {
5        "symbol": "TSLA",
6        "name": "Tesla Inc",
7        "type": "Equity",
8        "region": "United States",
9        "market_open": "09:30",
10       "market_close": "16:00",
11       "timezone": "UTC-04",
12       "currency": "USD",
13       "match_score": "0.8889"
14     },
15     {
16       "symbol": "TL0.DEX",
17       "name": "Tesla Inc",
18       "type": "Equity",
19       "region": "XETRA",
20       "market_open": "08:00",
21       "market_close": "20:00",
22       "timezone": "UTC+02",
23       "currency": "EUR",
24       "match_score": "0.7143"
25     },
26     ...
27     {
28       "symbol": "TL01.FRK",
29       "name": "TESLA INC. CDR DL-001",
30       "type": "Equity",
31       "region": "Frankfurt",
32       "market_open": "08:00",
33       "market_close": "20:00",
34       "timezone": "UTC+02",
35       "currency": "EUR",
36       "match_score": "0.3846"
37     }
38   ]
39 }
```

**Parameters:**

```
1  [
2    {
3      "Input Name": "keywords",
4      "Type": "String",
5      "Description": "Keywords to search, , e.g., company name, to retrieve
                       the ticker symbol for"
6    }
7  ]
```

### G.9 PYTHON

**Description:** Runs a python interpreter on a code snippet.

**Input Example:**

```
{
  "code": "print(4 + 5)"
}
```

**Output Example:**

```
{
  "result": "9",
  "error": ""
}
```

**Parameters:**

```
[
  {
    "Input Name": "code",
    "Type": "String",
    "Description": "The code snippet that we want to run on a python
        interpreter."
  }
]
```

### G.10 WOLFRAM ALPHA

**Description:** Accesses Wolfram Alpha to generate outputs from the Knowledgebase for computations and data queries. Wolfram Alpha excels at complex number-crunching, computation and calculations.

**Input Example:**

```
{
  "query": "what is Ronaldo's age?"
}
```

**Output Example:**

```
{
  "error": "",
  "result": "47 years 5 months 13 days"
}
```

**Parameters:**

```
[
  {
    "Input Name": "query",
    "Type": "String",
    "Description": "The query to perform computations/searches on. When
        unsure of your query search, try searching yourself on the
        website!"
  }
]
```

### G.11 GOOGLE SEARCH

**Description:** Performs a Google search and returns snippet results, without linked page details Google is often used for popular culture, location-awareness and crowdsourcing.

**Input Example:**

```
{
  "query": "What is the capital of France?",
  "location": "Paris"
}
```

**Output Example:**

```
{
  "error": "",
  "result": [
    {
      "source": "en.wikipedia.org",
      "date": "None",
      "title": "Paris",
      "snippet": "Paris is the capital and largest city of France. With
          an official estimated population of 2,102,650 residents as of 1
          January 2023 in an area of more than ...",
      "highlight": "Paris"
    },
    {
      "source": "home.adelphi.edu",
      "date": "None",
      "title": "Paris facts: the capital of France in history",
      "snippet": "Paris facts: Paris, the capital of France. Paris is the
          capital of France, the largest country of Europe with 550 000
          km2 (65 millions inhabitants).",
      "highlight": "Paris"
    },
    ...
    {
      "source": "britannica.com",
      "date": "None",
      "title": "France | History, Maps, Flag, Population, Cities, Capital
          , & ...",
      "snippet": "Get a special academic rate on Britannica Premium. The
          capital and by far the most important city of France is Paris,
          one of the world's preeminent cultural ...",
      "highlight": "Paris"
    },
  ]
}
```

**Parameters:**

```
[
  {
    "Input Name": "query",
    "Type": "String",
    "Description": "The search query."
  },
  {
    "Input Name": "location (Optional)",
    "Type": "String",
    "Description": "The geographical location for the search (optional)."
  }
]
```

