# OpenReview forum: "ToolComp: A Multi-Tool Reasoning & Process Supervision Benchmark"
_ICLR.cc/2025/Conference — Submitted to ICLR 2025_

### Official Review · Reviewer_nEha · 2024-11-01

**Soundness:** 2
**Presentation:** 3
**Contribution:** 2
**Rating:** 3
**Confidence:** 4

**Summary:**

The authors propose a new multiple tool-using benchmark named ToolComp. For each query, it not only labels the final answer similar to current benchmarks,  but also labels each reasoning/tool calling step for the whole problem solving procedure. ToolComp is developed through a collaboration between models and human annotators to ensure a accurate labelling.  The authors perform a comprehensive experiments to demonstrate the challenging nature of the dataset.

**Strengths:**

1. The authors perform comprehensive experiments on multiple LLMs families which may provides certain insights to the community.
2. With concerte implementation details, I could easily follow the whole paper to know how the work is actually done.

**Weaknesses:**

1.  I think the motivation for constructing this benchmark appears somewhat artificial: multi-step reasoning often requires
the evaluation for partial correctness or step-wise correctness of the reasoning trajectories. Why we need step-wise correctness? From the perspective of users relying on LLM for tool-using,  the primary concerns are always certain metrics: performance, cost, latency etc. The authors didn't provide a comprehensive justification for their motivation in the paper but with only few sentences in the introduction. This lack of clarity is a core issue within the study.

2. The labels utilized in this benchmark may not be accurate. Manually annotated ReAct trajectories do not necessarily serve as definitive golden labels. This benchmark seems to encourage LLM models to overfit specific ReAct reasoning chains rather than to develop correct reasoning paths. Defining the optimal trajectory is indeed challenging, so employing simpler yet meaningful metrics—such as accuracy, latency, and cost—would be more effective.

**Questions:**

Both ToolComp and ToolBench are evaluated through LLM with certain prompts and finally go through manual verfications.
Why you claim ToolBench doesn't have a verified final answer in Table 1?

---

> ### Author Response · Authors · 2024-11-20
>
> We thank the reviewer for useful comments and for appreciating the comprehensiveness of our evaluations. We address the reviewers questions and comments in the following sections. Additionally, we plan on incorporating all our discussions into the final version of the paper.
>
> **"I think the motivation for constructing this benchmark appears somewhat artificial: multi-step reasoning often requires the evaluation for partial correctness or step-wise correctness of the reasoning trajectories. Why we need step-wise correctness? From the perspective of users relying on LLM for tool-using, the primary concerns are always certain metrics: performance, cost, latency etc."**  This is an interesting question and you are correct that step-wise correctness is not the only thing that matters for function calling LLMs. This is exactly why we have created problems with unambiguous final answers and we present evaluations of leading LLMs on their ability to reach these final answers in Table 2.
>
> In addition to that critical component of the ToolComp benchmark, we also note that step-wise correctness is increasingly essential for LLMs, and we present three key motivations for its importance:
>
> 1. **Supporting Evidence from Research:** Recent studies by Lightman et al. and Wang et al. demonstrate the remarkable benefits of step-wise signals for improving reasoning in mathematical tasks [1, 2]. Similarly, Section 5 of this paper shows how incorporating step-wise learning signals significantly enhances the performance of reward models.
>
> 2. **Anecdotal Analogy:** Consider a math student tackling an Olympiad problem. There’s a difference between someone who solves 10% of the problem and someone who progresses 75% of the way. In the same vein, for LLM practitioners, benchmarking models based on their performance at various stages of reasoning—not just whether they ultimately arrive at the correct answer—provides much richer and more actionable insights.
>
> 3. **Empirical Results:** Figure 2 in this work explicitly illustrates a strong correlation between a model's ability to reason step by step (as evaluated via LLM as judge) and its overall accuracy in delivering correct final answers.
>
> Step-wise correctness, therefore, aligns with the core metrics of performance while enhancing interpretability and optimization in real-world applications.
>
> **"The labels utilized in this benchmark may not be accurate. Manually annotated ReAct trajectories do not necessarily serve as definitive golden labels. This benchmark seems to encourage LLM models to overfit specific ReAct reasoning chains rather than to develop correct reasoning paths."**:  We appreciate the reviewer’s concern, as we shared this exact consideration while developing the benchmark. To address this, we ensured that *neither the final answer evaluation nor the step-wise reasoning evaluation relies on full ReAct trajectories as definitive golden labels*. We apologize if this methodology was not sufficiently clear in the paper, and we would like to clarify how these evaluations are conducted:
>
> **Final Answer Evaluation.** In this evaluation, the model is assessed solely on whether it can produce the human-verified final answer for a given prompt. The model is free to choose any reasoning trajectory as long as it arrives at the correct final answer. This ensures that the evaluation is not conditioned on any predefined step-wise labels or specific ReAct trajectories.
>
> **Step-Wise Reasoning Evaluation.** This part of the evaluation tests the model's ability to reason about step-wise correctness by acting as a stepwise LLM-as-judge. Specifically, we have the model judge which of two possible next steps is more promising, an evaluation process which does not force it to follow or learn a specific fixed trajectory.
>
> Here’s how it works, broken down into the data collection and evaluation steps:
> 1. [Data collection] We use a generator model to create a reasoning path (called a "trajectory") that aims to reach the correct final answer.
> 2. [Data collection] If the generator makes any mistakes along the way, human annotators fix those mistakes. Each correction produces a pair of options: the original (incorrect) step and the corrected (better) step.
> 3. [Judge evaluation] These pairs are then used for evaluation. We use the model that is being evaluated to serve as an **LLM judge**, presenting it with two possible next steps at a specific point. The judge evaluates these options based on the prompt and the history of previous steps.
>
> This process ensures that the evaluation focuses on the model's ability to reason correctly at each individual step. It is not about memorizing or perfectly reproducing a specific fixed trajectory but instead about consistently choosing the most appropriate next move given the context.
>
> [1] Lightman, Hunter, et al. "Let's verify step by step."
>
> [2] Wang, Peiyi, et al. "Math-shepherd: A label-free step-by-step verifier for llms in mathematical reasoning."

---

> ### Author Response · Authors · 2024-11-20
>
> **"Both ToolComp and ToolBench are evaluated through LLM with certain prompts and finally go through manual verfications. Why you claim ToolBench doesn't have a verified final answer in Table 1?"** We would like to clarify that ToolComp is not evaluated through LLM with certain prompts followed by human verification. Rather, each ToolComp prompt is carefully crafted by human annotators to ensure that “*the final answer to the prompt can be programmatically verified*” (Lines 185–186). This means the correctness of the answer can be checked automatically using a well-defined procedure. Human annotators then use the tools to retrieve the ground truth final answer. Once the prompt and its corresponding final answer are established, we then use an LLM to evaluate the model’s final answer output. Specifically, the LLM compares the model’s final answer string to the programmatically verified ground truth string and checks for equality, similar to how you might use an LLM to compare two answers to a math question.
>
> In comparison, ToolBench by Qin et al. evaluates in two ways: “*Pass Rate: it calculates the proportion of successfully completing an instruction within limited budgets. The metric measures the executability of instructions for an LLM and can be seen as a basic requirement for ideal tool use; and (2) Win Rate: we provide an instruction and two solution paths to ChatGPT evaluator and obtain its preference (i.e., which one is better). We pre-define a set of criteria for both metrics and these criteria are organized as prompts for our ChatGPT evaluator. We evaluate multiple times based on ChatGPT to improve the reliability”. Then we calculate the average results from the evaluator* [3].” Pass rate simply grades the executability of a tool specified by the generator and win rate relies on ChatGPT to judge two trajectories in its entirety.
>
> In addition, for the other work that is also named ToolBench by Xu et al., 2 of the 8 tools are not evaluated by simply matching a verified final answer [4]. For instance, tool calls to the WebShop tool are awarded rewards computed using the WebShop environment, which uses “*a combination of programmatic functions that consider the attributes, type, options, and price of the chosen product.*” However, the remaining 6 out of the 8 tools evaluate the model’s final output matches with the verified ground truth. For this reason, we have decided to update Table 1 to give this ToolBench a checkmark for verified answers with an asterisk explaining our rationale.
>
> [3] Qin, Yujia, et al. "Toolllm: Facilitating large language models to master 16000+ real-world apis." arXiv preprint arXiv:2307.16789 (2023).
>
> [4] Xu, Qiantong, et al. "On the tool manipulation capability of open-source large language models." arXiv preprint arXiv:2305.16504 (2023).

---

> ### Author Response · Authors · 2024-11-26
>
> Dear Reviewer nEHa,
>
> We greatly appreciate your time and expertise in reviewing our work. We have provided detailed discussions in our response based on your valuable comments.
>
> As the discussion period nears its end, we would greatly appreciate any further input to ensure our rebuttal fully addresses your suggestions and strengthens our work.
>
> We thank you for your time and expertise.

---

> > ### Comment · Reviewer_nEha · 2024-11-30
> > **Thanks for your response**
> >
> > Thank you for your rebuttal! However, regarding my second point, I remain unconvinced by your explanation of the step-wise annotation process. It appears that the annotations are still dependent on ReAct, with the LLM-as-Judge mechanism functioning primarily as a filter to exclude incorrect trajectories at each individual step. From my perspective, this approach does not establish a true golden label.
> > Furthermore, assessing the correctness of each reasoning step should account for the context provided by subsequent steps, rather than evaluating each step in isolation. I decide to keep my score and suggest the authors to serious considering the annotation method and the motivation of this paper.

---

> > > ### Author Response · Authors · 2024-12-01
> > > **Response 1 of 2**
> > >
> > > We thank the reviewer and appreciate the opportunity to address these concerns in greater detail. We clarify the points raised, provide concrete examples to demonstrate the rigor, generality, and reusability of our approach, and address the misconceptions.
> > >
> > > **“..With the LLM-as-Judge mechanism functioning primarily as a filter to exclude incorrect trajectories at each individual step..”**
> > >
> > > This statement is not accurate. The LLM-as-Judge mechanism is used **solely as the evaluation method**, not during the data collection or trajectory correction process. During the data collection process, corrections to the model-generated (model is FireFunction-V1) steps are performed by **human annotators**, as clearly stated in both the paper and our previous rebuttal.
> > >
> > > To make this clearer, let’s walk through a **concrete example** to illustrate the data collection and evaluation process:
> > >
> > > ## Data Collection Process
> > >
> > > **Prompt**: *Using the 2023 birth-rate, death-rate, and population of Japan, assuming no one leaves or enters, what will be the projected population in 10 years?*
> > >
> > > Step 1:
> > >
> > > * Model Proposed Step:
> > >     - **Thought**: I need to use the google_search tool to calculate the population growth rate using the birth-rate, death-rate, and population of Japan in 2023.
> > >     - **Action**: google_search
> > >     - **Action Input**: Japan 2023 birth-rate per 1000
> > >     - **Observation**: ...6.6 births/1,000 population (2023 est.) Death rate, 11.7 deaths/1,000 population (2023 est.)...
> > > * Human Correction: None, the step is reasonable.
> > >
> > > Step 2:
> > >
> > > * Model Proposed Step:
> > >     - **Thought**: I need to use the python interpreter tool to calculate the population in 10 years using the population growth rate and the population of Japan in 2023.
> > >     - **Action**: python_interpreter
> > >     - **Action Input**: ```population = 120000000; population_growth_rate = 0.0012; population_in_10_years = population * (1 + population_growth_rate)**10; print(population_in_10_years)```
> > >     - **Observation**: 121447800.936
> > > * Human Correction: The model hallucinate the population value, hence the human corrected step is:
> > >     - **Thought**: I need to find the 2023 population of Japan using Google Search before proceeding.
> > >     - **Action**: google_search
> > >     - **Action Input**: Japan 2023 population.
> > >     - **Observation**: ...In 2023, the total population of Japan amounted to around 124.62 million inhabitants…
> > >
> > > Step 3:
> > >
> > > * Model Proposed Step:
> > >     - **Thought**: I need to use the python interpreter tool to calculate the population in 10 years using the birth and death rates and the 2023 population.
> > >     - **Action**: python_interpreter
> > >     - **Action Input**: ```population = 124620000; population_growth_rate = 0.0012; population_in_10_years = population * (1 + population_growth_rate)**10; print(population_in_10_years)```
> > >     - **Observation**: 126123541.272
> > >
> > > * Human Correction: The model does not take into account the death rate, hence the human corrected step is:
> > >     - **Thought**: I need to account for both birth and death rates when calculating the population in 10 years.
> > >     - **Action**: python_interpreter
> > >     - **Action Input**: ```population = 124620000; birth_rate_per_thousand = 6.6; death_rate_per_thousand = 11.7; total_births_extrapolated = birth_rate_per_thousand * (population / 1000); total_deaths_extrapolated = death_rate_per_thousand * (population / 1000); population_growth_rate = (total_births_extrapolated - total_deaths_extrapolated) / population; population_in_10_years = population * (1 + population_growth_rate)**10; print(population_in_10_years)```
> > >     - **Observation**: 118,408,275
> > >
> > > Step 4:
> > >
> > > * Model Proposed Step:
> > >     - **Thought**: I have enough information to answer the question.
> > >     - **Action**: finish
> > >     - **Action Input**: 118,408,275
> > >
> > > * Human Correction: None, the step is reasonable.
> > >
> > > ## Evaluation Using LLM-as-Judge
> > >
> > > Once the full dataset is collected, each model (Llama, Gemini, GPT, etc.) that we evaluate serves as an **LLM-as-Judge** to choose between an obviously wrong step and a human-corrected step. For the given example above, we would get two step-wise evaluation datapoints for the two occurrences of errors:
> > >
> > > Evaluation 1:
> > > - **Context**: Step 1
> > > - **Prompt/Choice**: We prompt the model to choose the better option between the model’s original Step 2 or the human-corrected Step 2.
> > > - **Correct Choice**: Human Step 2, as the model hallucinated Japan’s population.
> > >
> > > Evaluation 2:
> > > - **Context**: Step 1 and human-corrected Step 2.
> > > - **Prompt/Choice**: We prompt the model to choose the better option between the model’s original Step 3 or the human-corrected Step 3.
> > > - **Correct Choice**: Human Step 3, as the model ignored the death rate.
> > >
> > > This evaluation method ensures that models are judged on their ability to select the better reasoning step, grounded in the **actual context** of tool-use scenarios.

---

> ### Author Response · Authors · 2024-12-01
> **Response 2 of 2**
>
> **“...Annotations are still dependent on ReAct...”**
>
> ReAct is a general framework for formatting reasoning and tool calls, which has been **cited over 1800 times in the past 2 years** (which is more than any function calling format to our knowledge). Using a structured format for reasoning is essential for maintaining consistency in how the reasoning, tool selection, and tool inputs are represented and, most importantly, how the tool calls will be parsed. We chose ReAct precisely because it is the most canonical tool calling format, providing maximum generality across language models while imposing minimal constraints (merely that we prompt the LLM to reason before making a tool call). We would appreciate it if the reviewer could clarify why this commonly used format detracts from the merits of the benchmark itself. If you believe a different format would be better for a benchmark that is to serve as a general purpose test of the per-step reasoning abilities of LLMs on compositional tool use problems, can you please recommend a format that would A. generalize across LLMs, B. enable testing per-step reasoning abilities, and C. be reusable?
>
> **“...From my perspective, this approach does not establish a true golden label...”**
>
> The goal is not to create "true golden labels" for the step-wise annotations but rather to evaluate whether models can identify when clear errors in tool-use reasoning occur. Being able to judge correct versus erroneous trajectories requires reasoning about the multi-step task. As such, the use of human corrections ensures that at each step, the prompt has a correct and logical label, providing a reliable benchmark for step-wise reasoning evaluation. If we had evaluated step-wise accuracy by benchmarking against a “true golden label”, this approach would be flawed since there are potentially tens or hundreds of possible valid trajectories. We, however, are able to evaluate final-answer in this way, and each problem has a “true golden final answer”.
>
> **“...Assessing the correctness of each reasoning step should account for the context provided by subsequent steps...”**
>
> In practice, tool-augmented LLMs operate in a step-by-step manner, where decisions are made based on the current context of past actions and observations. Models do not roll out multiple trajectories and revise earlier steps based on future ones—this would be impractical and computationally expensive. Instead, we should be able to intercept inference when we can already recognize a mistake has occurred. Our evaluation framework mirrors the real-world operational context of tool-augmented LLMs, making it both realistic and meaningful.
>
> **“I decide to keep my score and suggest the authors to serious considering the annotation method and the motivation of this paper.”**
>
> We hope the above discussion clarifies the reviewer’s questions about  the annotation method and shows the rigour, generality, and reusability of the step-wise evaluations.
>
> As for the motivation of the paper, we appreciate the reviewer’s feedback, however, we respectfully disagree with the suggestion that the motivation of this paper has not been sufficiently justified. In our original rebuttal, we presented three detailed examples supporting the importance of step-wise reasoning: **Supporting Evidence from Research, Anecdotal Analogy, Empirical Results**.
>
> If the reviewer remains unconvinced by the motivations we have previously mentioned, we kindly request concrete counterarguments disputing these examples so that we can address them accordingly. Without such specific feedback, it is unclear what additional motivation the reviewer would like to see. Thank you again for taking the time to review our paper as these clarifications have made the paper stronger and will be worked into the final camera ready version.

---

### Official Review · Reviewer_AaPg · 2024-11-03

**Soundness:** 2
**Presentation:** 3
**Contribution:** 4
**Rating:** 5
**Confidence:** 4

**Summary:**

The paper introduces ToolComp, a comprehensive benchmark designed to evaluate multi-step tool-use reasoning capabilities of AI systems. ToolComp consists of 485 human-edited prompts that require chaining together multiple tool calls, accompanied by human-verified final answers and step-by-step process supervision labels. The authors evaluate the performance of various state-of-the-art models on ToolComp, assessing their ability to reach the correct final answer and their intermediate reasoning ability. Additionally, the paper compares the performance of process-supervised reward models (PRMs) and outcome-supervised reward models (ORMs) on ToolComp, demonstrating that PRMs generalize better than ORMs in improving complex tool-use reasoning. Key contributions include:

1. A comprehensive evaluation framework with both final answer and step-wise assessment
2. Evaluation of 16 models across 6 families on complex tool-use tasks
3. Analysis comparing process-supervised vs outcome-supervised reward models
4. Human-verified ground truth labels and process supervision annotations

**Strengths:**

1. ToolComp is a well-designed and comprehensive benchmark that addresses a gap in existing benchmarks by focusing on complex, multi-step tool-use reasoning.
2. The inclusion of human-verified process supervision labels is a strength, enabling a more nuanced evaluation of models' intermediate reasoning abilities.
3. The evaluation of various state-of-the-art models on ToolComp provides a clear understanding of the current landscape and challenges in this area.
4. The comparison between PRMs and ORMs is insightful and highlights the importance of process supervision in improving tool-use reasoning capabilities.

**Weaknesses:**

1. Limited analysis of tool interaction patterns and failure modes. Specially, the paper does not provide a detailed analysis of the specific failure modes or error patterns observed in the evaluated models, which could be valuable for further improving these models.
2. Limited discussion of benchmark biases and limitations.
3. Unclear scalability of process supervision approach.
4. Focus solely on text-based tasks, which may limit the generalizability of the findings to other modalities or domains.

**Questions:**

1. How representative are the synthetic training examples compared to human-verified ones?

2. What are the key patterns in how models chain different tools together?

3. How well does the process supervision approach scale to more complex tool combinations?

4. Could the authors elaborate on potential biases in the benchmark construction?

---

> ### Author Response · Authors · 2024-11-20
>
> We are thankful to the reviewer for their thoughtful review. We are pleased they find ToolComp well designed and comprehensive and they identify the importance of the process supervision labels. We address the reviewers questions and comments in the following sections. Additionally, we plan on incorporating our clarifications and discussion into the final version of the paper.
>
> **"...Specially, the paper does not provide a detailed analysis of the specific failure modes or error patterns observed in the evaluated models, which could be valuable for further improving these models."**: We thank the reviewer for raising this point. In the initial submission, we conducted a detailed analysis of failure modes and error patterns for the final answer evaluations. We refer the reviewer to **Appendix A.2**, which includes counts of common failure modes across all evaluated models.
>
> To provide even deeper insights, we have now extended this analysis to include failure modes and error patterns for the Step-Wise Reasoning Evaluation. We direct the reviewer to **Appendix A.3** for these updates. Specifically:
>
> - **Figures 6 and 7** break down the counts and distribution for substeps within the ReAct step that were incorrect whenever a model failed to reason about the better step.
> - **Figures 8 and 9** presents the counts and distribution for the normalized positions in the reasoning trajectory whenever a model fails to select the better step.
>
> Together, these figures highlight both where in the reasoning process the model struggles to identify the correct path and what types of errors are most common when misjudging (e.g., failures due to incorrect tool inputs). We hope this additional analysis offers valuable insights into the models’ reasoning capabilities and their specific areas for improvement.
>
> **"Focus solely on text-based tasks, which may limit the generalizability of the findings to other modalities or domains."** We appreciate this observation and acknowledge its relevance. This was a key consideration when designing ToolComp. As briefly mentioned in **lines 126–127**, we intentionally focus on text-only tasks to “*disentangle specialized capabilities and multi-step reasoning, allowing us to focus on the latter.*”
>
> Including additional modalities, such as vision or audio, could introduce failure modes stemming from a model’s lack of capability in that specific modality. This would make it harder to isolate and evaluate the models’ performance specifically on multi-tool, multi-step reasoning. By restricting our scope to text-based tasks, we ensure a more controlled environment for studying the reasoning process in detail.
>
> That said, we agree that extending this analysis to other modalities would be valuable future work to assess generalizability.
>
> **"How representative are the synthetic training examples compared to human-verified ones?"** The synthetic training examples are highly representative of the human-verified ones, as they are created using a process designed to closely mirror the human refinement pipeline. Here’s how we ensure this:
>
> **Consistent Seeding.** We use the same in-context examples that were initially employed to seed the prompts for creating ToolComp. This ensures a strong alignment in the generation process between synthetic and human-verified examples.
>
> **Filtering for Coherence and Verifiability.** To emulate the human refinement step, which ensures that each prompt is coherent and has a single programmatically verifiable answer, we generate an excess of candidate prompts. We then filter these prompts based on whether four state-of-the-art models (GPT-4o, GPT-4 Turbo, Claude 3.5 Sonnet, and Llama 3.1 70b) can arrive at the same final answer. This step ensures that the synthetic examples align with the rigor and clarity of human-verified examples.
>
> **Synthetic Process Supervision Labels.** For generating process supervision labels, we follow the exact methodology outlined in **Figure 1** for ToolComp, replacing human annotators with GPT-4o as the critic. GPT-4o is prompted with the same detailed instructions provided to human annotators, ensuring that the process supervision for synthetic examples is directly comparable to the human-verified labels.
>
> Through these steps, we replicate the rigor and structure of the human pipeline, making the synthetic training examples representative of the human-verified ones. These steps are described in more detail in **Appendix D.1**.

---

> ### Author Response · Authors · 2024-11-20
>
> **"What are the key patterns in how models chain different tools together?"** The reviewer raises an excellent question. Through extensive qualitative checks in the auditing stage, we observed several key patterns in how models chain different tools together. These patterns largely depend on the nature of the prompt and can be categorized as follows:
>
> **Parallel Tool Calls.** For prompts that require addressing multiple independent tasks or sub-questions, models often invoke tools in parallel. This pattern minimizes unnecessary dependencies and ensures efficient execution of tasks that can be completed independently.
>
> **Dependent Tool Calls.** When the task involves sequential reasoning or step-by-step problem-solving, models chain tools in a dependent manner, where the output of one tool informs the input to the next. Within this pattern, we noticed two sub-patterns:
> - **Internal Knowledge Utilization:** If the required intermediate information is reliably within the model’s internal knowledge, the model will often bypass making an initial tool call and directly provide the input for the subsequent tool.
> - **Tool-Dependent Reasoning:** When the necessary information is not within the model’s internal knowledge or is uncertain, the model will make an initial tool call to acquire the missing data before proceeding with subsequent steps.
>
> **Capability-Dependent Behavior.** More capable models, such as GPT-4o, are better at leveraging their internal knowledge and optimizing tool calls, avoiding unnecessary invocations when they can infer intermediate results. In contrast, less capable models, like Llama 3.1 8b, exhibit a heavier reliance on external tools, even for information that might be inferable from internal knowledge.
>
> These patterns illustrate how models balance efficiency and accuracy in tool chaining, adapting their strategies based on task requirements and their own capabilities. This insight also highlights the importance of evaluating models across diverse prompts to better understand their reasoning behaviors and tendencies.
>
> **"How well does the process supervision approach scale to more complex tool combinations?"** We thank the reviewer for raising this interesting point. To analyze how process supervision scales with complexity, we conducted an additional experiment comparing rank@1 performance between ORM and PRM across ToolComp prompts of varying complexity. Here’s the approach we used:
> Categorizing Prompt Complexity. We grouped ToolComp prompts into three categories—Easy, Medium, and Hard—based on the number of tool calls required in the human-verified trajectory to answer the prompt:
>
>  - **Easy:** Prompts solved in 1–4 steps.
>  - **Medium:** Prompts solved in 5–8 steps.
>  - **Hard:** Prompts solved in 9–12 steps.
>
> While there can be multiple valid trajectories of different lengths for the same prompt, we consider this categorization a reasonable proxy for complexity.
>
> **Findings and Insights.** **Figure 10 in Appendix B** (newly added) shows the performance comparison of ORM versus PRM across these categories. PRM consistently demonstrates better scalability for harder prompts, with the largest performance gains observed in the Hard category. This result highlights that PRMs are particularly effective in handling more complex queries requiring sophisticated reasoning across multiple tools.
>
> While this experiment offers valuable insights into the scalability of PRMs for complex prompts, we acknowledge that a broader study exploring how PRMs scale during training and inference with an increasing number of tools in the dataset would be a promising avenue for future research. We hope this experiment provides clarity on the scalability of process supervision approaches and appreciate the reviewer’s suggestion to explore this topic further.

---

> ### Author Response · Authors · 2024-11-20
>
> **"Could the authors elaborate on potential biases in the benchmark construction?"** We acknowledge that a discussion of potential biases in benchmark construction was missing from the paper and appreciate the opportunity to address it here. There are several biases to consider, some introduced by design and others inherent to the benchmark’s construction.
>
> **Human Bias in Step Correction.** During the step correction process, human annotators may favor certain tools that are more intuitive or convenient to use. This creates a skew in the Step-Wise Reasoning data, where some tools are overrepresented. However, this bias was deliberately allowed to ensure the dataset reflects how humans (and likely models) naturally prefer to use tools, optimizing for convenience and practicality.
>
> **Constraints for Unambiguous Prompts.** Each ToolComp prompt was designed to have an unambiguous, programmatically verifiable ground truth final answer. This constraint led to final output formats that are somewhat contrived, as they had to strictly adhere to the requirement for programmatic verifiability. While this ensures clarity in evaluation, it may limit the natural variability seen in real-world prompts.
>
> **Focus on Tool Composition.** Since the primary goal of ToolComp is to assess tool composition abilities, the benchmark does not include prompts that:
> - Require no tools.
> - Fall outside the scope of the provided tools.
> - Require clarifying questions or back-and-forth dialogue.
>
> These omissions mean the dataset does not encompass some real-life use cases but ensures focus on the core evaluation objective.
>
> **Bias in Generator Model.** Firefunction-v1, the base model used for generating initial trajectories for human annotators may introduce another layer of bias. The model may have preferences for certain tools or input formats, potentially influencing the dataset’s distribution.
>
> We appreciate the importance of acknowledging and addressing these biases and will include a detailed discussion in the final version of the paper.

---

> ### Author Response · Authors · 2024-11-26
>
> Dear Reviewer AaPg,
>
> We greatly appreciate your time and expertise in reviewing our work. We have provided additional failure mode analysis, further insights into the scalability of PRMs to complex tool use, and detailed discussions in our response based on your valuable comments.
>
> As the discussion period nears its end, we would greatly appreciate any further input to ensure our rebuttal fully addresses your suggestions and strengthens our work.
>
> We thank you for your time and expertise.

---

### Official Review · Reviewer_NrmG · 2024-11-08

**Soundness:** 3
**Presentation:** 4
**Contribution:** 3
**Rating:** 8
**Confidence:** 3

**Summary:**

This paper introduces ToolComp, a benchmark for evaluating LLMs on complex reasoning using multiple tools. Data from the benchmark is created semi-automatically. The prompts, intermediate steps and the final answers are first generated by the LLMs and then checked and revised by annotators. LLMs are then asked to 1) reason over the prompts for complex reasoning and to 2) select the human-corrected answer versus the model-generated incorrect answer from intermediate steps. The authors further study how multi-step tool-use in ReAct format benefits from using process reward models (PRM) versus outcome reward models (ORM), by training the RMs using additionally generated data.

**Strengths:**

1. The benchmark introduced in this paper, i.e. ToolComp, is novel, sound and solid.
- Novelty: They novelly benchmark LLM-as-a-judge on intermediate steps in multi-step tool use. (Cf. Table 1)
- Soundness: 1) The way they evaluate LLMs on multi-step tool use is sound and they also incorporate 95% CIs which makes the results more convincing. 2) The setup of Chat and Enterprise is reasonable. 3) The LLM-as-a-judge evaluation is sound and serves as an important benchmark for both LLM evaluation and PRM training (as the author study later in the paper).
- Solidness: The benchmark is heavily supervised, and the authors have done well-round study on the data and statistics of the benchmark.
2. The comprehensive study on PRM versus ORM in ReAct format 1) further strengthens the value of the LLM-as-a-judge evaluation 2) paves the way for future study on PRMs for multi-step tool use.
3. The paper is coherent, dense and fluent.

**Weaknesses:**

1. The author mentioned that ToolComp is complex. However, from Table 1 we can only know that it has fewer number of tools. It'll be more convincing if the authors can report quantitative comparisons, e.g., accuracy of LLMs, to other benchmarks. Or, it will also be more convincing if the authors can elaborate on why the other tools are not included.
2. As the authors mentioned in Table 2, the Llamas should be used with constrained decoding to guarantee valid outputs. It is unclear whether constrained decoding is still used in the experiments from Figure 3.
- If it's not the case, would the performance boost come from learning the correct output format (which might also be one of sources of the performance gain from SFT).
- If it is the case (still using constrained decoding), is it possible to study whether the PRM can still work without constrained decoding? That said, can PRM also provide supervision over format correctness? I think including more ablations can help readers and researchers better understand how and when PRMs that trained in this way work.

**Questions:**

1. I think the experiments in the paper are insightful, and the only question I want to ask is the one the authors mentioned in the limitation chapter:
**Is there any chance for the authors to train the PRM using human-generated process rewards?** A simple demonstration would be really helpful. But I know it would be either labor-intensive or lacking fair comparison if trained on the subset of ToolComp. Thus, I will not reduce my score if the authors could not address the question.

---

> ### Author Response · Authors · 2024-11-20
>
> We thank the reviewer for their thorough review and we are pleased that they find the work novel, sound, and comprehensive. In the following sections, we address the questions and comments raised by the reviewer. We plan on incorporating all our discussions into the final version of this paper.
>
> **"The author mentioned that ToolComp is complex. However, from Table 1 we can only know that it has fewer number of tools…"** The reviewer brings up a good point regarding complexity and we acknowledge its relevance. Complexity can be defined in multiple ways. Since ToolComp is constructed to test planning abilities, when we use the word complexity, we refer to the planning and tool composition complexity. We refer the reviewer to Figure 11 in the Appendix, which plots a histogram of the trajectory lengths of the human supervised trajectory. We see that most of the prompts required between 4-8 steps. While we acknowledge that these trajectories are single valid paths, this distribution provides a proxy for the complexity of the prompts with regards to planning and chaining tool calls together.
>
> **"...Or, it will also be more convincing if the authors can elaborate on why the other tools are not included."** We appreciate the opportunity to elaborate on this decision. Our findings show that adding more tools significantly increased annotation complexity, which in turn led to a decrease in the quality of examples to include in the benchmark. Specifically, when more tools were available, annotators often created prompts that increased complexity in an undesirable way—by asking disjointed questions that required multiple tools but lacked meaningful compositional reasoning.
>
> For example, instead of crafting prompts that required the output of one tool to be used as input to another (true multi-tool, multi-step reasoning), annotators tended to create tasks that simply listed unrelated sub-questions requiring different tools. This undermined the primary goal of the benchmark: to assess and enhance models' ability to plan and chain tools in a coherent, interdependent manner. By limiting the number of tools, we found that annotators were better able to focus on composing prompts that require thoughtful planning and complex reasoning. This resulted in higher-quality examples that align more closely with the benchmark's objectives.
>
> Another key consideration when designing ToolComp was adding more tools that require additional modality. As briefly mentioned in lines 126–127, we intentionally focus on text-only tasks to “*disentangle specialized capabilities and multi-step reasoning, allowing us to focus on the latter.*” Including additional modalities, such as vision or audio, could introduce failure modes stemming from a model’s lack of capability in that specific modality. This would make it harder to isolate and evaluate the models’ performance specifically on multi-tool, multi-step reasoning. By restricting our scope to text-based tasks, we ensure a more controlled environment for studying the reasoning process in detail. That said, we agree that extending this analysis to other modalities would be valuable future work to assess generalizability.
>
> **"It is unclear whether constrained decoding is still used in the experiments from Figure 3."** Constrained decoding is not used for the experiments in Figure 3/Section 5.
>
> **"If it's not the case, would the performance boost come from learning the correct output format (which might also be one of sources of the performance gain from SFT)."** We agree with the reviewer's judgment in that the greedy sampling pass@1 performance boost from SFT is likely to both better adherence to ReAct format and better reasoning about tools. However, in the PRM versus ORM experiments, we ensure that the performance comparison is not influenced by output format errors. Specifically, we sample trajectories from the generator model (Base Llama-3.1-8b-Instruct and Fine-tuned Llama-3.1-8b-Instruct) in such a way that only trajectories reaching a final answer without any mistakes in the output format are considered. This is achieved through iterative retries until we obtain 50 format-compliant trajectories per problem. As a result, the trajectories used to evaluate both PRM and ORM are free from issues related to ReAct format adherence, ensuring a fair comparison based solely on scoring promising trajectories higher and the supervision type.

---

> ### Author Response · Authors · 2024-11-20
>
> **"Is there any chance for the authors to train the PRM using human-generated process rewards?"** We share the reviewer’s enthusiasm for exploring the use of human-generated process supervision labels, particularly given the significant performance gains observed with synthetically generated training data. However, gathering a sufficiently large dataset of high-quality human annotations for training is a time-intensive process and was not feasible within the timeline of this work.
>
> That said, we agree that this is a promising direction for future research, and we are excited to explore how incorporating human-generated process rewards might further enhance performance and provide deeper insights into model behavior. We look forward to pursuing this in follow-up studies.

---

> > ### Comment · Reviewer_NrmG · 2024-11-24
> >
> > Thank you for your clarification. I will keep my score at 8.

---

> > > ### Author Response · Authors · 2024-11-26
> > >
> > > We appreciate your response!

---

### Official Review · Reviewer_8N3E · 2024-11-08

**Soundness:** 3
**Presentation:** 3
**Contribution:** 3
**Rating:** 6
**Confidence:** 3

**Summary:**

This paper introduces a new benchmark aimed at evaluating models’ ability to use multiple tools in multi-step tasks. The task prompts and golden answers in this benchmark are generated collaboratively by both LLMs and human annotators. Each instance is manually inspected to ensure task feasibility and answer accuracy, underscoring the rigor of the dataset. Additionally, the authors train two types of reward models, ORM and PRM, to assess their effectiveness in enhancing model performance on tool-usage tasks, demonstrating through experiments that PRM outperforms ORM.

**Strengths:**

1. The motivation for this paper is strong, addressing a gap in the field by providing a dataset that supervises the process of tool usage rather than just the final outcome, allowing for accurate evaluation of multi-step tool-use reasoning.
2. The method of generating queries and answers combines LLM outputs with human validation and careful inspection of each instance in the dataset, ensuring high efficiency.
3. For dataset evaluation, the authors implement LLM grading to avoid the limitations of exact-match grading, which can evaluate some complex answers.
4. The authors present a thorough analysis demonstrating the strong correlation between step-wise reasoning accuracy and final answer accuracy, and they provide additional evidence through reward model training that process rewards are critical for this task.

**Weaknesses:**

1. The evaluation is limited to general LLMs, without testing LLMs specifically fine-tuned for tool use. Including specialized tool-use LLMs could yield deeper insights into model performance in this particular task domain and highlight differences in tool-specific reasoning abilities.
2. Figure 8 (line 1206) reveals that among the 11 tools included, only four are frequently used, while the rest appear in very few instances. This usage pattern, particularly in the ToolComp-Chat subset, may introduce biases that limit generalizability. Additional analysis of tool usage distribution in the ToolComp-Enterprise subset could provide a more balanced view and potentially address these biases.

**Questions:**

1. In Section 3.2 (Prompt Creation) and Appendix A.1, the authors mention that initial IC examples are crafted by human annotators and subsequently expanded with examples generated by GPT-4 Turbo. Given this approach, could reliance on GPT-4 Turbo alone in prompt generation lead to a model-specific skill bias in these tasks? Would diversifying prompt generation across several LLMs reduce this risk and potentially improve generalization?
2. The ToolComp-Enterprise set has around 50% more instances than ToolComp-Chat, though Chat is designed for broader, general-purpose applications. Since general use cases are often more prevalent in real-world scenarios, wouldn’t it make sense for the Chat set to have more examples to reflect typical model usage?

**Details Of Ethics Concerns:**

No ethics concerns.

---

> ### Author Response · Authors · 2024-11-20
>
> We thank the reviewer for their insightful and useful comments. We appreciate that the reviewer acknowledges the strong motivation of the work, and finds our benchmark construction and evaluation highly efficient, extensive, and rigorous. In the following sections, we address the questions and comments raised by the reviewer. We plan on incorporating all our discussions into the final version of this paper.
>
> **"The evaluation is limited to general LLMs, without testing LLMs specifically fine-tuned for tool use."** With the ever increasing popularity of Tool Use LLMs, most frontier models are specifically fine-tuned for Tool Use. We refer the reviewer to the following papers and links that explicitly mention that these frontier models have been specifically fine-tuned for tool use:
> - GPT Family: https://openai.com/index/function-calling-and-other-api-updates/
> - LLama Family: https://arxiv.org/pdf/2407.21783 (Section 4.3.5)
> - Gemini Family: https://arxiv.org/pdf/2312.11805 (Section 6.5.2)
> - Claude Family: https://docs.anthropic.com/en/docs/build-with-claude/tool-use
> - Cohere Family: https://cohere.com/blog/tool-use-with-command-r
>
> "**Including specialized tool-use LLMs could yield deeper insights into model performance**" We acknowledge that there are additional models that are further fine-tuned specifically for Tool Use. We evaluate 2 such models, namely, Team-ACE/ToolACE-8B (Finetuned on LLaMA-3.1-8B-Instruct) and meetkai/functionary-medium-v3.1 (Fine Tuned on Llama-3.1-70B-Instruct) [1, 2]. The following table summarizes their performance on final answer accuracy and step-wise reasoning:
>
> ## Final Accuracy Eval
>
> | Model                   | Total               | Chat               | Enterprise         |
> |-------------------------|---------------------|--------------------|--------------------|
> | Toolace-8b             | $4.54 \pm 1.49$    | $2.53 \pm 1.71$   | $5.92 \pm 2.22$   |
> | Functionary-medium-v3.1 | $29.13 \pm 4.04$   | $27.91 \pm 6.26$  | $29.96 \pm 5.30$  |
>
> ## Step Wise Evals
>
> | Model                   | Total               | ReAct              | Action Plan        |
> |-------------------------|---------------------|--------------------|--------------------|
> | Toolace-8b             | $60.48 \pm 2.31$   | $63.04 \pm 2.66$  | $53.40 \pm 4.58$  |
> | Functionary-medium-v3.1 | $65.38 \pm 2.25$   | $67.36 \pm 2.58$  | $59.89 \pm 4.50$  |
>
> Overall, we find that these models do not perform as well as their base counterparts. While Toolace-8b shows a slight improvement in step-wise reasoning evaluations compared to Llama-3.1-8b-Instruct, its significantly lower performance on final answer evaluations can be attributed to its reluctance to tackle questions requiring creative tool use.
> For instance (out of many instances), given the prompt:
>
> "*Using the 2023 birth-rate, death-rate, and population of Japan, assuming no one leaves or enters, what will be the projected population in 10 years?*"
>
> and the tools *Google Search* and *Python Interpreter*, the model responds:
>
> "*The given question lacks the parameters required by the available functions. The functions provided are for searching Google and running Python code, but they do not support demographic projections or calculations based on birth-rate, death-rate, and population data.*"
>
> This response demonstrates a failure to creatively integrate the tools to compute an answer, despite the question being solvable within the provided framework.
>
> For Functionary-medium-v3.1, there is no single failure mode that stands out. However, its weaker step-wise reasoning abilities may explain its overall performance drop relative to Llama-3.1-70b-Instruct in final answer evaluations. This suggests that consistent deficiencies in reasoning across intermediate steps compound to negatively affect the final output quality.
>
> "**Given this approach, could reliance on GPT-4 Turbo alone in prompt generation lead to a model-specific skill bias in these tasks. Would diversifying prompt generation across several LLMs reduce this risk and potentially improve generalization?**" We acknowledge that while relying on one model may lead to some model-specific skill bias, in practice we believe the intervention and collaboration with expert annotators greatly reduced the opportunity for model-specific bias. During the last step (Step 4: Human Refinement) of prompt creation, there were relatively many more human refinements to increase prompt coherence and complexity. The instructions to the annotators were to adjust the prompts to increase their complexity and coherence while maintaining the original set of target tools for the prompt. Moreover, we note that the top models for Gemini, GPT, and Claude families perform around the same for the final answer accuracy, suggesting that there is no practical advantage for the GPT model family.
>
> [1] Liu, Weiwen, et al. "ToolACE: Winning the Points of LLM Function Calling."
>
> [2] https://github.com/MeetKai/functionary

---

> ### Author Response · Authors · 2024-11-20
>
> "**Figure 8 (line 1206) reveals that among the 11 tools included, only four are frequently used, while the rest appear in very few instances…**" For the construction of the benchmark, the initial prompts were seeded such that every tool would be used equally in the ToolComp-Enterprise setting, but in practice certain tools ended up being more convenient and powerful. For example, Wolfram Alpha was able to recall certain stock prices and weather, Google Search was also able to search up weather, stock symbols, stock prices, etc. In addition, we chose not to bias/force annotators to use certain tools in order to enforce diversity. This decision was made to reflect real life practices of matching the true distribution of humans using these tools. Moreover, this is also why we did not treat these as golden trajectories, rather we provide one good path to the correct solution. This also reinforces our decision for pairwise LLM as judge evaluation for the step wise critiques, instead of point-wise evaluations.
>
> **"...This usage pattern, particularly in the ToolComp-Chat subset, may introduce biases that limit generalizability."**  You raise a good point here. The ToolComp-Chat subset indeed utilizes fewer tools and that’s why we have also created the ToolComp-Enterprise subset. The tools we include in ToolComp-Chat (Google Search and Python Interpreter) are valuable and useful tools, occurring frequently in many practical applications. In particular these two tools are common in leading user-facing chatbots such as ChatGPT and Claude. We specifically designed this subset with fewer tools in order to better test this common chatbot setting. For ToolComp-Enterprise, our prompt creation process requires LLMs to use other tools, but even in this setting it is in general to do more with more powerful tools (eg math can also be executed in the more powerful python tool and not only the calculator tool), so we still see some bias in tool usage, but we believe that this skew is reflective of real use cases in which more general tools will be used more often. That said there are many questions where the best or only way to solve a problem is to use a specific tool, such as for queries requiring historical weather information.
>
> "**...Additional analysis of tool usage distribution in the ToolComp-Enterprise subset could provide a more balanced view and potentially address these biases."** We agree with this feedback and have added the tool usage distribution for ToolComp-Enterprise and ToolComp-Chat separately. We refer the reviewer to Figures 15 and 16 in the Appendix. In addition, to get a sense of how one tool was used in place of another, the following provides some crude statistics on when Google Search and Wolfram Alpha was used in place of another relevant tool in the ToolComp-Enterprise subset:
>
> Google Search was used for:
> - Weather 43 times
> - Stock information 22 times
> - Wiki articles 89 times
>
> Wolfram was used for:
> - Weather 13 times
> - Stock information 9 times
>
> "**The ToolComp-Enterprise set has around 50% more instances than ToolComp-Chat, though Chat is designed for broader, general-purpose applications. Since general use cases are often more prevalent in real-world scenarios, wouldn’t it make sense for the Chat set to have more examples to reflect typical model usage?**" As tool-use capabilities continue to gain traction, model development is increasingly focused on creating versatile agents capable of leveraging a wide range of tools to enhance their functionality. Given this trend, we believe ToolComp-Enterprise is becoming, or will soon become, more representative of typical model usage. Moreover, ToolComp-Enterprise includes 11 tools compared to ToolComp-Chat's 2 tools, which naturally necessitates a larger number of prompts to adequately cover the broader and more diverse toolset.

---

> > ### Comment · Reviewer_8N3E · 2024-11-21
> >
> > Thank you for your explanation! I will keep my rating at 6.

---

> > > ### Author Response · Authors · 2024-11-26
> > >
> > > Thank you for your response!

---

### Official Review · Reviewer_kidg · 2024-11-09

**Soundness:** 3
**Presentation:** 3
**Contribution:** 3
**Rating:** 5
**Confidence:** 4

**Summary:**

The paper proposes a new dataset, ToolComp for complex planning involving the use of 2 or more tools. The dataset involves the use of 11 different tool calls across 485 different prompt and final answers. The final dataset consists of 1731 per step level annotations. The authors extensively compare the performance of instruction tuned models on the process supervision label prediction, as well as performance improvement in using PRMs vs ORMs when training to use as Reward Models.

**Strengths:**

> Tool use is an important area, and having more dataset/benchmark contributions in this area is useful. The use of step level supervision/verification also is proving to be a promising area making this contribution pretty timely.

> The empirical comparisons in this paper with existing IT checkpoints in zero-shot evals are pretty extensive and thorough

**Weaknesses:**

> The contributed dataset is relatively small. With just 485 prompts, and 1731 per step level annotations. It makes it hard to justify using this dataset for large scale model training and pragmatically can only be used as a benchmark.

> The experiments in section 5 lack sufficient details. I had to go into the details of the experiments in the Appendix to understand Section 5 better.

**Questions:**

> Fig 3 compares Base vs SFT on verification using PRMs vs ORM. It is unclear if here was the Llama 3.1 Instruct the base model used for sampling trajectories or the final reward model.

> Is the synthetic data mentioned as training data in Section 5 and Appendix E part of the dataset that is released with this paper? Why / why not?

---

> ### Author Response · Authors · 2024-11-20
>
> We are grateful to the reviewer for their thoughtful review. We were pleased to see they acknowledged the importance of this work, the promise in the avenue of step level supervision/verification, and found our extensive evaluations of current models rigorous and insightful. We address the reviewers questions and comments in the following sections. Additionally, we plan on incorporating all our clarifications and discussion into the final version of the paper.
>
> **"With just 485 prompts, and 1731 per step level annotations. It makes it hard to justify using this dataset for large scale model training and pragmatically can only be used as a benchmark."** We apologize if this was not clear in the paper, but the 485 prompts with human verified final answers and the 1731 per step level annotations make up our ToolComp benchmark, which is designed **solely** for benchmarking and **not for training purposes**. The synthetic training dataset introduced in Section 5 is completely separate from the ToolComp benchmark and is only meant to serve as a preliminary study into outcome-supervised versus process-supervised signals. We use the ToolComp benchmark as an evaluation set to thoroughly evaluate the performance of the two approaches.
>
> **"The experiments in section 5 lack sufficient details…"**: We appreciate the reviewer’s feedback regarding the level of detail in Section 5. Many of the specific details for the experiments in this section are indeed provided in the **Appendix D**, and in the main paper, we have aimed to reference the relevant sections of the Appendix wherever applicable. To address this concern and enhance clarity, we have made revisions to improve the presentation and flow of the text in Section 5, ensuring that key details are more comprehensible. If there are still aspects that remain unclear after these edits, please let us know which parts and we will be more than happy to provide additional clarification or elaborate further.
>
> **"[In] Fig 3…It is unclear if here was the Llama 3.1 Instruct the base model used for sampling trajectories or the final reward model."**:  In alignment with ORM versus PRM evaluations by Lightman et al and Wang et al, we use a generator model to generate 50 trajectories per problem in ToolComp (ToolComp is used only for evaluation here) and then use the corresponding reward models (ORM or PRM) to select the highest scoring trajectory [1, 2]. In Figure 3, we plot the results for two different generators. On the left, the generator is the base Llama-3.1-8b-Instruct model. On the right, the generator is a fine-tuned Llama-3.1-8b-Instruct model that was fine-tuned on just the preferred chains of the synthetic training dataset. We additionally want to clarify that Llama-3.1-8b-Instruct weights are used to initialize both the ORM and PRM and once they are trained on the synthetic training dataset, the reward models serve just to score (and not sample) the generated trajectories; we have made this detail more clear in **Section 5.1 - Evaluation** and **Appendix D.5**. Once we score each trajectory, the final answer of the highest scoring trajectory per problem is compared with the corresponding human-verified ground truth final answer to measure accuracy. Please let us know if you have any further questions and we will be happy to address them.
>
> **"Is the synthetic data mentioned as training data in Section 5 and Appendix E part of the dataset that is released with this paper? Why / why not?"**: We thank the reviewer for inquiring about the synthetic training dataset. This allowed us to reflect on the potential benefit of releasing this training data and we have decided to release it as part of this work! The synthetic training data includes 11K prompts, 11K preferred and dis-preferred full trajectories for ORM training, and 15K preferred and dis-preferred steps for PRM training. With the release of the synthetic training dataset, we also plan on releasing the evaluation pipeline that ToolComp uses.  We believe these resources will help facilitate further research in exploring the effectiveness of process-supervised signals for complex tool-use reasoning. We have included this release as part of the list of key contributions on **lines 87-90**.
>
>
> [1] Lightman, Hunter, et al. "Let's verify step by step." arXiv preprint arXiv:2305.20050 (2023).
>
> [2] Wang, Peiyi, et al. "Math-shepherd: A label-free step-by-step verifier for llms in mathematical reasoning." arXiv preprint arXiv:2312.08935 (2023).

---

> ### Author Response · Authors · 2024-11-26
>
> Dear Reviewer kidg,
>
> Thank you for your time and expertise in reviewing our work. In response to your valuable feedback, we have improved the clarity of our work, committed to releasing the synthetic training data, and provided detailed discussions in our response.
>
> As the discussion period nears its end, we would greatly appreciate any further input to ensure our rebuttal fully addresses your suggestions and strengthens our work.
>
> We thank you for your time and expertise.

---

### Author Response · Authors · 2024-11-20

We sincerely thank the reviewers for their valuable and constructive feedback. We are delighted that the reviewers found our work to be well-written, novel, and insightful in advancing the evaluation of tool use in language models, as well as in assessing the performance of process supervision versus outcome supervision.

In response to the excellent feedback, we have made the following updates to enhance our work:

1. **Step-Wise Failure Analysis**: We conducted an in-depth analysis of step-wise failures, which is now detailed in *Section A.3*.
2. **Complexity Scaling Experiment**: We added a new experiment comparing PRM and ORM performance scaling with increasing prompt complexity, presented in *Section B*.
3. **Release of Additional Resources**: We will release the full synthetic process supervised training data, which includes 11K prompts, 11K preferred and dis-preferred full trajectories for ORM training, and 15K preferred and dis-preferred steps for PRM training. We will also release the ToolComp evaluation pipeline to encourage further research into compositional tool use, agents, and process supervision.

We are grateful for the reviewers’ insights, which have greatly contributed to improving the quality and impact of this work.

---

### Meta-Review · Area_Chair_XPTt · 2024-12-19

**Metareview:**

The paper "ToolComp: A Multi-Tool Reasoning & Process Supervision Benchmark" introduces a novel benchmark, ToolComp, designed to evaluate the multi-step tool-use reasoning capabilities of large language models (LLMs). The benchmark, developed collaboratively with human annotators and LLMs, includes 485 human-edited prompts and corresponding final answers, along with step-by-step process supervision labels. The authors benchmark various state-of-the-art LLMs on ToolComp, demonstrating its challenging nature, and compare the effectiveness of process-supervised reward models (PRMs) versus outcome-supervised reward models (ORMs) in improving tool-use reasoning. Their findings underscore the superior performance of PRMs, particularly in handling complex queries.

#### Contributions
The primary contributions of the paper are:
1. **ToolComp Benchmark**: A novel, high-quality benchmark for evaluating multi-step tool-use reasoning in LLMs, with both final answer and step-wise process supervision labels.
2. **Empirical Evaluation**: Comprehensive assessment of 16 LLMs across six families, highlighting the challenges in tool-use reasoning.
3. **PRM vs. ORM Analysis**: Insightful comparison demonstrating the advantages of process supervision, particularly for complex tasks.
4. **Additional Resources**: Commitment to release synthetic training data and the ToolComp evaluation pipeline, facilitating further research.


#### Weaknesses
1. **Limited Scale of Benchmark**: With 485 prompts, ToolComp is smaller compared to some existing benchmarks, limiting its use for large-scale training. However, the authors clarify that ToolComp is designed primarily for benchmarking, not training, and this limitation is mitigated by the benchmark's high-quality design.

2. **Focus on Text-Based Tasks**: The benchmark's restriction to text-based tasks may limit its generalizability to other modalities. While the authors justify this choice to focus on multi-step reasoning without modality-specific challenges, extending the benchmark to include other modalities could enhance its applicability.

3. **Potential Biases**: Reviewers raised concerns about potential biases in the benchmark construction, such as tool usage skew and reliance on specific models for prompt generation. The authors acknowledged and discussed these biases, which are inherent to such benchmarks but do not undermine the overall contribution.

4. **Unclear Scalability of PRMs**: While the paper demonstrates PRMs' superiority, reviewers questioned how well they scale with more complex tool combinations. The authors conducted additional experiments to address this, but scalability remains an area for future exploration.

5. **Misunderstandings in Reviewer Feedback**: Some reviewers misunderstood aspects of the methodology, such as the role of LLM-as-Judge in step-wise evaluations. While the authors provided clear rebuttals, these misunderstandings may have influenced the reviewers' scores.

**Additional Comments On Reviewer Discussion:**

- **Reviewer kidg**:
  - **Concerns**: Small dataset size, lack of detail in experiments, unclear use of base models in PRM/ORM evaluations.
  - **Authors' Responses**: Clarified benchmark purpose, improved experiment presentation, detailed PRM/ORM methodology, and committed to releasing synthetic training data.
  - **Outcome**: The reviewer maintained their original score (marginally below acceptance threshold), despite the authors' thorough responses.

- **Reviewer 8N3E**:
  - **Concerns**: Lack of evaluation on tool-specific LLMs, potential biases in tool usage distribution, reliance on a single model for prompt generation.
  - **Authors' Responses**: Provided evaluations of tool-specific LLMs, analyzed tool usage distributions, and justified the use of GPT-4 Turbo with human refinements.
  - **Outcome**: The reviewer kept their score (marginally above acceptance threshold), indicating satisfaction with the responses.

- **Reviewer NrmG**:
  - **Concerns**: Lack of quantitative comparisons to other benchmarks, unclear use of constrained decoding, potential for human-generated process rewards.
  - **Authors' Responses**: Clarified complexity metrics, explained constrained decoding exclusion, and discussed the feasibility of human-generated rewards.
  - **Outcome**: The reviewer maintained a high score (accept, good paper), affirming the paper's value.

- **Reviewer AaPg**:
  - **Concerns**: Limited failure mode analysis, potential biases in benchmark construction, unclear PRM scalability.
  - **Authors' Responses**: Provided detailed failure mode analyses, discussed potential biases, and conducted PRM scalability experiments.
  - **Outcome**: The reviewer did not respond to the rebuttal, but the authors requested re-evaluation based on their comprehensive responses.

- **Reviewer nEha**:
  - **Concerns**: Unclear motivation for step-wise correctness, potential overfitting to ReAct trajectories, inaccuracies in benchmark comparisons.
  - **Authors' Responses**: Provided detailed justifications for step-wise correctness, clarified evaluation methodology, and corrected benchmark comparisons.
  - **Outcome**: The reviewer remained unconvinced about step-wise annotations and maintained a low score (reject, not good enough), despite the authors' thorough rebuttals.


The paper presents contribution to the field of tool-augmented LLMs by introducing ToolComp, a well-designed benchmark that focuses on multi-step reasoning and process supervision. The authors have demonstrated a high level of responsiveness to reviewer feedback, conducting additional experiments and providing clear clarifications. However, some limitations, such as the benchmark's scale and modality focus, remain after rebuttal. I suggest authors take the reviewers' comments seriously and improve the paper.

---

### Decision · Program_Chairs · 2025-01-22

Reject